

# Luminescence properties and dating of proglacial sediments from northern Switzerland

Daniela Mueller[1], Frank Preusser[1], Marius W. Buechi[2], Lukas Gegg[2], Gaudenz Deplazes[3]

[1]Institute of Earth and Environmental Sciences, University of Freiburg, 79104 Freiburg, Germany
[2]Institute of Geological Sciences, University of Bern, 3012 Bern, Switzerland
[3]NAGRA - Nationale Genossenschaft für die Lagerung radioaktiver Abfälle, 5430 Wettingen, Switzerland

*Correspondence to*: Daniela Mueller (daniela.mueller@geologie.uni-freiburg.de)

**Abstract.** Luminescence dating has become a pillar of the understanding of Pleistocene glacial advances in the northern foreland of the Swiss Alps. However, both quartz and feldspar from the region are equally challenging as dosimeters with anomalous fading, partial bleaching and unstable components being some of the obstacles to overcome. In this study, luminescence properties of coarse- and fine-grained quartz, feldspar and polymineral fractions of eight samples from a palaeovalley, Rinikerfeld, in northern Switzerland are assessed and found appropriate for dating. While anomalous fading of
the IRSL signal of the feldspar and polymineral fraction is observed, fading corrected coarse-grained feldspar ages are consistent with those derived from quartz. In general, coarse-grained quartz and feldspar as well as the fine-grained polymineral fractions are in agreement and present negligible evidence for partial bleaching. However, ages derived from fine-grained quartz are found to underestimate those of the coarse-grained quartz fractions. Impact of total dose rate on finite ages was assessed but age underestimation is likely due to grain size dependent luminescence properties. The top six samples indicate
sedimentation of at least 16.6 m during Marine Isotope Stage 6 with a rapid transition from a lacustrine environment to a landscape dominated by colluvial deposits. For the two lowest samples, no finite ages are derived.

## 1 Introduction

Pleistocene glaciations in the northern foreland of the Swiss Alps have been studied since the early 19[th] century (summarised in Preusser *et al.*, 2011). Whereas it appears that a minimum of eight glacial advances have shaped the lowlands, consensus
on the exact number and timing is still lacking. In the last three decades, numerical dating turned into a crucial component in the reconstruction of the environmental past of northern Switzerland (e.g. Preusser, 1999; Graf *et al.*, 2007; Kock *et al.*, 2009; Dehnert *et al.*, 2012). In particular, luminescence dating has become a pillar of chrono-stratigraphy and understanding of glacial advances (Preusser *et al.*, 2011). However, luminescence dating of glacial and pro-glacial deposits can be complex (e.g. Duller, 1994, 2006; Spencer and Owen, 2004). For one, in such environments sediment sources and sinks are often within
short distance of each other and transport may occur in turbid waters which both reduces the chance for grains to experience sufficient sunlight exposure to reset or bleach any pre-existing luminescence signals. This effect will consequently lead to age overestimations and requires consideration in such deposits. The measurement of single grains or small aliquots has been advised to allow for monitoring of luminescence signal commonly presented as equivalent dose ($D_e$) distributions and the



isolation of a proportion of the $D_e$ distributions that is considered well bleached (e.g. Olley *et al.*, 2004; Duller, 2006;
Trauerstein *et al.*, 2017). The luminescence signal of quartz is more readily bleachable than this of feldspar (Godfrey-Smith *et al.*, 1988; Murray *et al.*, 2012), and is hence favoured as dosimeter (e.g. Lowick *et al.*, 2015). But, to asses partial bleaching of samples from northern Switzerland, Trauerstein *et al.* (2017) have recommended to compare luminescence signals obtained from both dosimeters. However, quartz and feldspar from the region have found to be equally challenging for dating.

For quartz from the wider region, unstable components were reported for some (e.g. Klasen *et al.*, 2016) but not all samples
(e.g. Gaar *et al.*, 2013). Moreover, laboratory dose response curves for quartz were often best fitted with a double saturating exponential function that accounts for high dose responses beyond single exponential behaviour (Lowick *et al.*, 2010; Dehnert *et al.*, 2012). This phenomenon is also known for samples from other regions where luminescence ages are either in agreement (e.g. Murray *et al.*, 2008; Pawley *et al.*, 2010) or disagreement (e.g. Lai, 2010; Timar *et al.*, 2010; Timar-Gabor *et al.* 2011) with independent age control. However, the physical reason for this behaviour has yet to be identified (Wintle, 2008). In
northern Switzerland, quartz luminescence ages of up to ca. 250 ka have been found reliable (Anselmetti *et al.*, 2010; Dehnert *et al.*, 2012; Lowick *et al.*, 2015; Buechi *et al.*, 2017), but not beyond. Hence, this approach is not suitable to establish an independent chronology for the recently modified Mid Pleistocene glaciation history (Graf, 2009; Preusser *et al.*, 2011). significantly limiting the use of quartz as a dosimeter for the region.

While Feldspar requires longer for bleaching, it is known to saturate at higher doses and therefore often allows to date much
older deposits than quartz (cf. Duller, 1997). Yet, the infrared stimulated luminescence signal of feldspar measured at 50 °C (IRSL) often suffers from anomalous fading contributing to age underestimation (Wintle, 1973; Spooner, 1994). Different approaches to determine and to account for the loss of signal over time have been proposed (Huntley and Lamothe, 2001; Auclaire *et al.*, 2003; Lamothe *et al.*, 2003; Huntley, 2006; Kars *et al.*, 2008). These approaches rely on observing signal loss over hours to days within laboratory experiments and deducing from this signal loss that has occurred over geological time
scales. In such storage tests, fading has been observed for most samples from northern Switzerland (g-values between 1 and 3 % per decade). However, uncorrected IRSL $D_e$ values were either (1) beyond the linear part of the dose response curve and therefore unsuited for most fading correction approaches, (2) corrected IRSL $D_e$ values were close to saturation and consequently rejected for age determination or (3) uncorrected feldspar ages were in better agreement with quartz ages and therefore favoured (Dehnert *et al.*, 2012; Lowick *et al.*, 2012, 2015; Gaar and Preusser, 2012; Buechi *et al.*, 2017). Only Gaar
*et al.* (2013) found correcting for fading appropriate and necessary for samples <100 ka. For dating of older samples, Lowick *et al.* (2012) tested alternative measurement protocols that target signals that are more stable than the IRSL signal, the post infrared infrared stimulated luminescence signals (pIRIR). The pIRIR signals are measured at higher temperatures after an initial readout of IRSL signals at 50 °C (Thomsen *et al.*, 2008; Buylaert *et al.*, 2011). While fading is expected to decrease or become negligible for the more stable pIRIR signals, these take longer to bleach by sunlight during transport. For the used
$pIRIR_{225}$ and $pIRIR_{290}$ signals, fading was still observed and age overestimations led to the conclusion that these approaches are not beneficial for the investigated waterlain-sediments from northern Switzerland (Lowick *et al.*, 2012).





Here, luminescence properties of samples from the Rinikerfeld in the northern foreland of the Swiss Alps are assessed. Scientific drilling was conducted at a stratigraphic key site within the former glacier forefield as part of a larger campaign. The campaign is aimed to acquire further insights into the long-term glacial and fluvial landscape evolution of Northern
Switzerland. Modelling of future erosion scenarios will be based on the gained knowledge and thereby assist in the assessment of the safest and most suitable location for a prospective Swiss nuclear waste depository. For this, the establishment of a chrono-stratigraphy is essential and, hence, dating is a crucial component. Therefore, feldspar, polymineral as well as fine- and coarse-grained quartz are investigated to exploit the dating potential of the samples from this site. Performance tests, signal intensity and composition of quartz as well as fading properties of feldspar are discussed and two approaches to correct for
fading are applied. Eventually ages are derived and discussed with regard to problems found for other studies of the region.

## 2 Material and method

### 2.1 Site setting and samples

The study site (Fig. 1) is located within the Rinikerfeld, at the eastern tail of the Jura Mountains, about 30 km NW of Zurich.
Rinikerfeld, is part of an extensive palaeovalley structure and situated in an elevated position, ca. 50 m above the nearby present-day Lower Aare Valley. At the site, the palaeovalley is carved into Mesozoic bedrock (Bitterli-Dreher *et al.*, 2007) and presumed to be of Mid-Pleistocene age (Bitterli-Dreher *et al.*, 2007; Graf, 2009). It was spared from glacial overprint during the Last Glacial Maximum (Bini *et al.*, 2009) and about 40 m of Quaternary sediments were recovered in cores within a scientific drilling campaign (Gegg *et al.*, 2018). The composite core consists of about 4 m of glacial diamicton overlain by
ca. 2 m sandy gravel that gradually transition into ca. 24 m of lacustrine clay and silt. The upper lacustrine unit progressively coarsens and is capped by ca. 7.5 m of diamicts and colluvium (Fig. 2).

For luminescence dating, samples were obtained from ca. 10 cm in diameter cores recovered in plastic liners (Table 1); three samples were taken from the gravelly to sandy diamicts unit (RIN1, 2, 3), one sample of lacustrine sands (RIN4) and one of gravelly sand and silt (RIN13). The two latter embrace the lacustrine unit whereof another three samples were taken (RIN5, 6,
8). Sample material was collected in ca. 10 cm segments whereby the outer rind and the surface, exposed to daylight during core cutting and splitting, were allocated for dose rate determination and the inner core was used for $D_e$ determination.

### 2.2 Sample preparation and measurement

To determine $D_e$ values, samples were wet-sieved and treated with HCl and $H_2O_2$ to remove carbonates and organic matter,
respectively. Of five samples (RIN1 to RIN4, RIN13), coarse grained (150-200 and 200 to 250 µm) potassium-rich feldspar (F) and quartz (Q) were separated using sodium polytungstate at densities of 2.58 g cm$^{-3}$ and 2.7 g cm$^{-3}$. The outer rind of the quartz grains was etched in 40 % HF for 1 h followed by a 32 % HCl treatment for another hour to eliminate any fluoride



precipitates. For the remaining three samples (RIN5, 6, 8) and RIN13, the fine-grained fraction (4-11 µm) was separated using settling under Stokes' Law. One half of the sample remained pristine as polymineral fraction (PM) while the other half was

treated with hexafluorosilicic acid for seven days in order to obtain purified quartz (fQ). The fine-grained fractions were suspended in acetone and settled onto cups of 9.8 mm in diameter (>1.1*10$^6$ grains). For the coarse-grained fractions, sample material was mounted onto cups using silicon oil with diameters of 1 mm for feldspar and 4 mm for quartz (ca. 13 and 200 grains, respectively).

$D_e$ values were obtained with a Freiberg Instruments *Lexsyg Research* using LEDs with peak emission at 458 nm (BSL) for

quartz and at 850 nm (IRSL) for feldspar and polymineral. BSL and IRSL signals were detected by an ET9235QB photomultiplier tube filtered through a 3 mm KG 3 Schott glass combined with a 5 mm BP 365/50 Delta interference filter or a 3 mm BG 39 Schott glass and a 3.5 mm HC 414/46-1 AHF Brightline interference filter (referred to as "410 nm-filter combination), respectively. Additionally, a cardboard barrier was mounted into the filter wheel for IRSL measurements (for details see 3.3). Laboratory irradiation was given by a $^{90}$Sr/$^{90}$Y beta source mounted into the reader. The beta source was

calibrated using *Risø* calibration quartz batches 108 (4-11 µm) and 118 (180-250 µm) to ~0.10 and 0.11 Gy s$^{-1}$, respectively. Notable is the preheat behaviour of the used *Lexsyg Research* reader (Lexsyg ID 09-0020); at a heating rate of 5 °C s$^{-1}$, the so-called 110 °C TL peak emerges at much lower temperatures than expected. For example, for *Risø* calibration quartz (batch 118) TL counts are peaking at $87 \pm 3$ °C during preheat. This is in agreement to observations made by Schmidt *et al.* (2018b) who compared the preheat behaviour of various OSL readers and discovered the appearance of the 110 °C TL peak within a

range of 60 °C. This has to be considered when assessing appropriate measurement protocols (see 3.1).

Performance tests were conducted on both the coarse- and fine-grained fractions of two representative samples (RIN2Q, RIN2F, RIN5fQ, RIN5PM). Preheat plateau tests were carried out on the natural dose while for thermal transfer and dose recovery tests, sample material was exposed to an daylight lamp for 16 h (Q, fQ) to 30 h (F, PM). All performance tests were conducted using preheating previous to the natural, regenerative and test doses for which the aliquots were heated with 5 °C s$^{-}$

$^1$ to the tested temperature and held for 10 s (Q, fQ) or 60 s (F, PM).

Both BSL and IRSL measurements were conducted for $D_e$ determination following the SAR protocol (Murray and Wintle, 2000) with test doses of ~46 Gy and ~23 Gy administered, respectively. The initial signal was derived from the first 0.4 s of the BSL and the first 15 s of the IRSL signals. A late background subtraction using the last 40 s (BSL) or 50 s (IRSL) was applied. $D_e$ values were calculated using the numOSL package for R (Peng *et al.*, 2018).

For Q and fQ, a combined recycling and IR depletion ratio (>20 %) step was implemented at the end of each sequence to check the adequacy of the sensitivity correction and for feldspar contamination. A maximum of four aliquots per sample failed this rejection criterion. None of the measured Q, fQ, F and PM aliquots presented dim (<3*BG level) or imprecise (>20 %) test dose signals and were hence accepted. For RIN3 F, one aliquot was rejected as it failed the recycling ratio test (>20 %) and for RIN13 Q, three aliquots were rejected as their $D_e$ values were in saturation.

For dose rate determination (Table 2), sample material was dried and radio-nuclide content was determined via high-resolution gamma spectrometry at VKTA (Dresden, Germany). A mean alpha efficiency factor of $0.05 \pm 0.01$ (a-value) was assumed for





all fQ (Buechi *et al.*, 2017) as well as for the F and PM fractions (Preusser, 1999). For the two latter an internal potassium content of $12.5 \pm 0.5$ % was used (Huntley and Baril, 1997; Gaar *et al.*, 2013). The cosmic contribution was determined for present day depths following Prescott and Hutton (1994). Water content relative to the dry weight of the sample and capacity

of water absorption (DIN 18132:2012-04, 2016) were determined in the laboratory. Representative long-term water content estimates of 20 to $25 \pm 5$ % were used for total dose rate ($D_{total}$) determination. Age calculations were conducted with ADELE 2017 software ([www.add-ideas.com](www.add-ideas.com); Degering and Degering, 2020).

## 3 Results

### 3.1 Performance test

For preheat plateau tests on the natural signal of Q (RIN2), statistically consistent (1 σ) $D_e$ values were obtained on the larger grain size fraction for preheats between 200 and 240 °C (Fig. 3). $D_e$ values of fQ (RIN5) for this preheat temperature range are consistent at 2 σ (Fig. 3). A given dose of ~130 Gy was fully recovered from both tested samples (RIN2 Q and RIN5 fQ; Fig. 3) at 240 °C with measured-to-given-dose ratios (M/G-ratios) of $1.01 \pm 0.03$ and $1.05 \pm 0.04$, respectively. Maximal thermal

transfer was detected with <4 Gy which equals to less than 3.5 % of the natural $D_e$ and is therefore considered negligible. Hence, a preheat temperature of 240 °C was chosen for both grain size fractions of quartz (Q, fQ) for all investigated samples. For F, poor recovery was obtained for preheats above 190 °C. Considering the poor performance at high temperatures and the early emergence of the 110 °C TL peak on the used *Lexsyg Research* reader (see 2.2), lower temperatures were tested and a full recovery of the given dose (~130 Gy) was possible for F and PM with a preheat of 170 °C. No plateau but increasing

trends of $D_e$ values from the natural signal were observed between 150 and 210 °C for F and between 170 and 210 °C for PM. Natural $D_e$ values decrease rapidly for preheats above 210 °C while thermal transfer behaves inversely. However, thermal transfer is with <2 Gy (~1.5 % of the natural $D_e$) negligible. To investigate whether thermal quenching induces an underestimation of $D_e$ values at high preheat temperatures, preheat measurements were conducted on aliquots of RIN2 F following Wallinga *et al.* (2000). Therefore, the natural signal was bleached with IR LEDs, a fixed dose of ca. 90 Gy was

administered and each aliquot was preheated to 50 °C for 10 s followed by IRSL at 50 °C for 0.1 s. This was repeated for preheat temperatures between 50 and 300 °C in 25 °C steps. After the highest preheat, the entire measurement sequence was repeated to allow for normalisation (Fig. 4). An increase in normalised IRSL sensitivity would be expected if the electron trapping probability changes with temperature. Yet, a decrease was observed for temperatures above 200 °C suggesting that it is not a change in trapping probability causing the lower natural $D_e$ values and dose recovery ratios. Rather the high preheat

temperature may cause a removal of signal commonly used for $D_e$ determination. Therefore, a preheat temperature of below 200 °C is most advisable. Best performance of the SAR protocol was obtained at a preheat of 170 °C (Fig. 3) which suggests a much lower preheat than conventionally used. Buylaert *et al.* (2011) advice against the use of preheats below 200 °C due to thermally unstable signals induced during laboratory irradiation. However, their study was conducted on a *Risø* reader with





different preheat temperature behaviour. It is stressed that temperature parameters of *Risø* and *Lexsyg* readers are not readily
comparable. Nevertheless, to investigate a potential contribution from artificially filled unstable traps, short shine pulse
annealing tests were conducted on aliquots of RIN2 F with naturally and laboratory irradiated material following Wallinga *et al.* (2000). Each aliquot was preheated for 10 s respectively and then stimulated with IR LEDs at 50 °C for 0.1 s. Preheats were
conducted consecutive in 10 °C steps from 130 to 400 °C. The signal was normalised with a test dose measurement of the
same aliquot and renormalized to the first measurement (Fig. 5). Signal distribution of both the natural and laboratory induced
irradiation are indistinguishable and it is unlikely that a thermally unstable component from artificially filled traps contributes
to the IRSL signal even at low preheat temperatures. Thermal stability of IRSL signals from the here investigated samples is
given and thermal quenching was not detected, therefore, preheats of 170 °C are considered appropriate. In absence of preheat
plateaus of the natural signals, the dose recovery test results are indicative (RIN2 F $0.97 \pm 0.03$; RIN5 PM $0.98 \pm 0.03$) and
hence, preheats of 170 °C were used all for measurements.

Yet, the differences in temperature parameters make it necessary to test whether the configuration of the *Lexsyg Research*
reader effects the temperature during stimulation as for preheating. Therefore, $D_e$ values of the natural signal were obtained at
stimulation temperatures of 30, 50 and 70 °C for one representative sample (RIN2 F) (Fig. 6). While the scatter in the natural
$D_e$ values decreases with stimulation temperature, mean $D_e$ values of all stimulation temperatures are statistically consistent
with each other, implying that differences between the reader systems may not be as pronounced in the low temperature range
as they are at temperatures above 150 °C.

### 3.2 Quartz OSL signal properties

Quartz grains from the investigated site are relatively dim and rarely inherit luminescent behaviour. Tests with an EMCCD
camera showed that <1 % emitted a signal when stimulated with BSL. This is common for quartz from Switzerland
(Trauerstein *et al.*, 2017) and to allow for bright enough signals during measurement, an aliquot size of 4 mm in diameter was
chosen. While this allows for ca. 200 grains to be measured at once, only few rains are expected to significantly contribute to
the emitted luminescence signal and, thereby, these are considered as small aliquots.

The signals emitted by Q and fQ are dominated by the fast component and reduce to background after about 15 to 20 s of
stimulation (Fig. 7). To further investigate the BSL signals, signal decomposition was conducted (Fig. 8) using the numOSL
package in R (Peng *et al.*, 2018) that applies the 'Levenberg-Marquardt algorithm' suggested by Bluszcz and Adamiec (2006).
The initial signals consist of 95 % for Q and 90 % for fQ of the fast component while the relative contribution of the medium
components to the natural and regenerative doses are almost identical. The latter is considered an indicator for a stable medium
component (Steffen *et al.*, 2009). Further, when assessing $D_e(t)$-plots with 0.4 s intervals of Q from RIN2 (Fig. 7), mainly
consistent $D_e$ values are obtained for the measurement of the natural dose as well as for the dose recovery experiment which
indicates uniformity of signal contribution. For RIN5 fQ, $D_e$ values decrease with the shift of integration intervals between 0
and 2 s stimulation time. However, this phenomenon is only present in the natural signal but not in those of dose recovery





tests. Li and Li (2006a) observed a similar decrease of $D_e$ values within the first 3.6 s for their DGF-1 sample (coarse-grained, Chinse, aeolian deposit) and found an explanation in a thermally unstable medium component. They proposed the use of $D_e(t)$-plots to derive $D_e$ values separately from the fast and medium component by fitting their Equation (3). For RIN5 fQ, negligible

difference were found for $D_e$ values derived using either component assuming photo-ionising cross sections as proposed by Jain *et al.* (2003) and Li and Li (2006b). However, Steffen *et al.* (2009) found that the equation is highly dependent on the photo-ionising cross section of the individual components, the determination of which introduces an array of uncertainties and was therefore deemed unpractical.

$D_e$ values derived using early background subtraction (0.4 to 1.0 s; cf. Cunningham and Wallinga, 2010) are statistically

consistent with those obtained using late background subtraction (2 σ). This indicates an unproblematic slow component. For $D_e$ determination of Q and fQ, the first 0.4 s of the BSL signal were found appropriate as initial signal and a late background subtraction using the last 40 s was applied. An example of an extended dose response curve is shown for one aliquot Q from RIN 2 in Fig. 9. Laboratory saturation levels exceed 600 Gy (Q) and 800 Gy (FQ) and $2*D_0$ values are below 400 Gy (Q) and 580 Gy (fQ; Table 1). Dose response curves of both Q and fQ are well fitted with a double saturating exponential as well as a

single saturating exponential function. The latter implies that a single type of trap is responsible for the signal (Aitken, 1998).

### 3.3 Feldspar IRSL signal properties

In contrast to Q, the natural signal of 1 mm F aliquots (ca. 13 grains) is very bright and cause saturation of the photomultiplier tube, in particular at high doses. Noteworthy is that only few grains inherit an effective luminescence signal with even less being particularly bright as shown by tests with an EMCCD camera (Fig. 11). A Schott NG-11 neutral density filter was tested

together with the 410 nm-filter combination but failed in protecting the photomultiplier tube from saturation. A cardboard disc with a hole in its centre was mounted into the filter wheel, reducing the photon passage to 4 mm in diameter which allowed to retrieve a detectable signal. A dose recovery test (given dose of ca. 130 Gy) was conducted with and without using the cardboard disc (Fig. 10). Results of both setups are statistically consistent with each other and suggest that the use of the

cardboard is unlikely to impact the reliability of the $D_e$ determination. Hence, the cardboard was used in all IRSL measurements within this study.

$D_e(t)$-plots with 1.5 s intervals show consistent results over the first 200 s of the IRSL signal as presented for both the natural signal and the artificial signal induced during dose recovery tests of RIN2 F and RIN5 PM (Fig. 7). For age determination, $D_e$ values were derived using the first 1.5 s of the signal and a late background subtraction of the last 50 s. An example of an

extended dose response curve is shown for one aliquot from RIN2 F in Fig. 9. Laboratory saturation levels exceed 1500 Gy (F) and 1800 Gy (PM) and $2*D0$ values are about 600 Gy (F) and above 425 Gy (PM; Table 1). For both F and PM dose response curves are well fitted with both a double saturating exponential and a single saturating exponential function.





### 3.4 $D_e$ distributions and ages

The $D_e$ distributions of the four top Q samples show the tendency towards being positively skewed and have overdispersion (OD) values between $21 \pm 3$ and $25 \pm 3$ % (Fig. 12). Arnold *et al.* (2007) recommend the application of a MAM3 (Galbraith *et al.*, 1999) for $D_e$ distributions of partially bleached samples that are significantly positively skewed and/or show OD values of >40 %. Following this, skewness of RIN1, RIN2 and RIN4 require for the use of a MAM3. However, the decision tree of Arnold et al. (2007) is based on fluvial samples with divers bleaching histories that are comparably young (<20 ka). For older

samples, it is suggested that distributions may vary as time-dependent factors are expected to contribute to the data spread (Galbraith and Roberts, 2012). For example, the impact of beta microdosimetry (Mayya *et al.*, 2006) will increase with the age of the sample. Therefore, the application of a MAM3 was refrained from and derived CAM $D_e$ values of between $144.2 \pm 6.8$ and $194.7 \pm 7.8$ Gy were used for the age determination of the top four samples (Table 1). Q ages for these samples range between $163.5 \pm 9.1$ (RIN4) and $179.6 \pm 11.0$ ka (RIN3) and are statistically consistent with each other (1 σ).

The $D_e$ distributions of RIN13 Q is also positively skewed but presents a much higher OD value, $56 \pm 10$ %, in comparison to the four top Q samples. the measured aliquots, 12 % are in saturation only allowing for the assessment of a truncated distribution. In this case, the application of the MAM3 is also recommended (Arnold *et al.*, 2007). Besides an expected impact of beta microdosimetry, fitting uncertainties are likely accountable for the skewness and spread of the distribution. For RIN13 Q, the natural doses are projected on the high proportion of the dose response curves emphasising any fitting uncertainty.

Therefore, a MAM3 $D_e$ value is unlikely to represent the distribution most appropriately and the CAM $D_e$ value is considered to represent a more conclusive lower limit of the distribution. The CAM $D_e$ value of $255.4 \pm 34.2$ Gy is consistent with the $2*D_0$ limit of $274.1 \pm 16.3$ Gy and it is arguable whether an obtained age at this range is still reliable. However, the minimum CAM age of $185.8 \pm 25.6$ ka is consistent with ages of the top four samples and from this and the stratigraphy it can be deduced that RIN13 is of similar age or older than RIN1 to RIN4.

The four fQ samples (RIN5, RIN6, RIN8, RIN13), representing depths of 10 to 36 m, show normally distributed $D_e$ values (Fig. 2, Fig. 12, Table 1). This is to be expected for fine-grained samples as the number of grains on each aliquot will have an averaging effect that will mask any skewness of the $D_e$ distributions. CAM $D_e$ values of $195.0 \pm 4.4$ to $388.0 \pm 12.0$ Gy were derived which are all well below $2*D_0$ (Table 1). For the lowest samples (RIN13), the obtained age RIN13 with $216.8 \pm 12.8$ ka is close to the upper limit of reliable quartz ages in the region (Lowick *et al.*, 2015). Obtained fQ ages for the two top most

lacustrine samples are $108.0 \pm 5.6$ ka (RIN5) and $113.5 \pm 5.9$ ka (RIN6). These ages are between 30 and 40 % lower than ages derived for the Q fractions of RIN1 to RIN4. This is from a chrono-stratigraphy perspective unreasonable and can be ascribed to either issues with the $D_e$ value or the dose rate determination. The latter will be discussed in detail in section 4.2. However, if fQ ages are equal or older than Q ages higher $D_e$ values are expected for fQ. This is due to higher $D_{total}$ given for fine-grained sediments. But, $D_e$ values of RIN4 and RIN5 are almost equal (ratio $1.00 \pm 0.05$) resulting in a much lower age of RIN5 with

a negative offset to RIN4 by $56 \pm 11$ ka.





None of the F and PM $D_e$ distributions show a trend towards being skewed and OD values are between 0 and 17 % (Fig. 12). This is to be expected for the fine-grain PM aliquots that each represent a $D_e$ value derived from over one million grains. For F, each aliquot consists of only about 13 grains and the $D_e$ distributions are not affected by skewness. The CAM was applied for all F and PM samples following the decision tree of Arnold *et al.* (2007). CAM $D_e$ values of between 140.9 ± 3.1 (RIN1

F) and 412.8 ± 14.1 Gy (RIN13 F) were derived for the F fraction and between 148.2 ± 1.6 Gy and 332.7 ± 7.2 Gy for the PM fraction (Table 1). The offset between RIN4 F and RIN5 PM is with 19 ± 3 Gy less pronounced than in the quartz fractions.

### 3.5 Fading attributes

As the IRSL signal commonly is subject to a loss of signal over time, fading tests were conducted following Preusser *et al.*
(2014) for a given dose of ~130 Gy with storage times of up to 10 h per aliquot on the sample arm. This reduces the possibility of sample material being lost during mechanical transfer of the aliquot between sample arm and storage carousel. For all fading tests, aliquots previously used for $D_e$ determination were utilised. Two fading correction procedures were conducted, following Lamothe *et al.* (2003) and Kars *et al.* (2008), respectively, using the R Luminescence package (Kreutzer and Mercier, 2019; King and Burow, 2019).

Fading properties (ρ'-value of 2.21*10⁻⁶, g-value of 3.1 ± 0.6 % per decade) were obtained for RIN2 as a representative sample for the F fractions and fading correction was applied to the F measurements, providing corrected $D_e$ values and ages (Table 1). Following Kars *et al.* (2008), the corrected F $D_e$ values, except for RIN13 F, are about 16 % higher than those corrected using Lamothe *et al.* (2003). 'Kars' corrected $D_e$ values are between 247.2 ± 6.5 Gy (RIN1) and 297.3 ± 5.5 Gy (RIN4) and well below 2*$D_0$ (RIN1 560 ± 6 Gy, RIN2 603 ± 8 Gy, RIN3 590 ± 8 Gy, RIN4 612 ± 24 Gy). For RIN13 F, a significant number

of aliquots 12 (Lamothe correction) and 19 ('Kars' correction) out of 30 measured samples are in saturation. Minimum $D_e$ values of >694.5 and >973.7 Gy are obtained, respectively, which are beyond the 2*$D_0$ limit (650.5 ± 31.8 Gy). However, for all other samples the 'Kars' corrected F ages are in good agreement with those of the Q fraction (statistically consistent at 1 σ, Table 1). It cannot be excluded that this effect is due to an overestimation of both 'Kars' corrected F and Q ages. Due to the lack of evidence for the prior, the Kars *et al.* (2008) fading correction approach is considered most appropriate for samples of

this study.

For PM, fading properties (ρ'-value of 1.89*10⁻⁶, g-value of 2.4 ± 0.6 % per decade) were obtained from one representative polymineral sample (RIN5). Following Lamothe *et al.* (2003), the corrected ages are 20 to 100 % lower than those corrected using Kars *et al.* (2008). As the latter have shown good agreement for F and Q ages, ages calculated using 'Kars' corrected PM $D_e$ values are presented in the following. Corrected PM ages of RIN5 and RIN6 are, with 138.4 ± 6.6 and 169.0 ± 8.9 ka,

28 and 50 % higher than the associated fQ ages. While RIN5 is younger than to be expected from the Q and fading corrected F ages of the overlaying samples (RIN1 to RIN4), the derived fading corrected age is still consistent at 2 σ with the latter. The corrected PM age of RIN6 is statistically consistent (at 1 σ) with the ages of the overlaying samples. Consequently, an age



offset as observed between Q and fQ cannot be reported for F and PM and confidence of the chosen 'Kars' fading correction is given.

For RIN8 PM and RIN13 PM, corrected ages of $471.2 \pm 47.1$ and $525.8 \pm 53.2$ ka were calculated, respectively. Fading corrected $D_e$ values are with $1100.5 \pm 96.7$ Gy (RIN8) and $955.4 \pm 85.6$ Gy (RIN13) far beyond the according $2*D_0$ values (RIN8 $480.1 \pm 39.5$ Gy, RIN13 $514.5 \pm 38.6$ Gy). For RIN13 PM, this is in agreement with the minimum age of >436.6 ka obtained for the F fraction of the same sample. There are three possible explanations for the high $D_e$ values of RIN8 and RIN13; 1) the sediment was deposited about half a million years ago, 2) the applied fading correction overestimates for doses

derived from the high dose range of the response curve or 3) these $D_e$ values are the result of an averaged signal with a greater contribution of partially bleached, high residual signals. The latter is an effect that is to be expected for glacier-proximal deposits that are likely to have had limited light exposure during short transport times. Samples RIN8 and RIN13 are possibly from such an environment. However, no drastic increase in $D_e$ values was observed for the fQ fraction, but the OSL signal of quartz bleaches faster than feldspar. Nevertheless, corrected $D_e$ values are higher than the $2*D_0$ values. This measure has been

introduced as a dating limit for quartz (Wintle and Murray, 2006) but was found to be as well an appropriate measure for adequately dating feldspars (Zhang and Li, 2020). Therefore, it has been refrained from using the presented results for finite age determination of samples RIN8 and RIN13.

## 4 Discussion

### 4.1 Fading correction

One fading property, the g-value, determined for this study is in good agreement with those obtained for comparable studies in the investigated region (2-3 % per decade; Lowick *et al.*, 2012, 2015; Gaar *et al.*, 2013; Buechi *et al.*, 2017). However, most studies have abstained from relying on fading correction for several reasons. Buechi *et al.* (2017) concluded that already uncorrected PM IRSL ages exceeded the reliable range for dating. Lowick *et al.* (2012) found that corrected IRSL ages

overestimated what was to be expected from independent age controls while uncorrected IRSL ages were in good agreement. In Lowick *et al.* (2015), fading correction was rejected as most corrected IRSL ages were higher than quartz ages obtained for the same samples. Gaar and Preusser (2012) found lower g-values of between 0.5 and 1.9 % and argued that corrected PM ages lead to overestimation and much higher ages than those derived for fQ. In contrast, Gaar *et al.* (2013) recommended fading correction for one well-bleached sample as corrected F and Q ages were consistent with each other. However, those

studies used a fading correction following Huntley and Lamothe (2001), a correction based on the loss of luminescence signal measured on artificially irradiated aliquots and extrapolated to a decadal percentage (g-value) and designed for natural values that are projected onto the linear part of the dose response curve. For most investigated samples, this is not the case and two different approaches to account for anomalous loss of signal over time were applied here. Lamothe *et al.* (2003) found a dose rate correction equation that incorporates differences of irradiation in nature and those of the laboratory irradiation source.





This equation is also applicable for geologically old samples. Kars *et al.* (2008) published an approach that requires the estimation of the sample specific density of recombination centres (ρ'-value). This approach may allow correction beyond the linear part. Therefore, these two fading correction measures were found most appropriate for the here investigated samples and were tested. However, 'Kars' corrected F ages have been found to be in good agreement with those of the Q fraction (see 3.5, Table 1) and are therefore considered most appropriate for samples of this study.


## 4.2 Age comparison

For the top four samples, Q and fading corrected F ages range between 159.6 ± 6.6 (RIN4) and 179.6 ± 11.0 ka (RIN3) and are, thereby, statistically consistent with each other at 2 σ. For the two samples (RIN5, RIN6) below, a discrepancy between fQ and PM ages (10 to 30 %) was reported. Also, fQ ages of these two samples are up to 40 % lower than those of the overlying

samples RIN1 to RIN4. This leads to an age offset of ca. 35 ka between Q and fQ that is unaccountable from a chrono-stratigraphic perspective. The observed age offset equals 30 to 40 % difference and is likely to be emphasised through the actual age calculation which in terms relies onto derived $D_{total}$ values. The used $D_{total}$ values are all based on the assumptions that firstly, (1) the long-term average estimates of the water content are accurate, and that secondly, (2) used alpha efficiency values are appropriate in this context. However, in contrast to the top four coarse-grained samples (RIN 1 to RIN4), $D_e$ values

of RIN5 and RIN6 were determined on the fine-fraction. Therefore, (3) the effect of grain-size dependent luminescence characteristics has to be assessed as well.

### 4.2.1 Water content and age determination

A water content of 20 ± 5 % was assumed for all but two samples which is within 5 % uncertainty of the measured field water

content (Table 2). RIN5 and RIN6 have a slightly higher field water content (23 to 24 %) due to their argillaceous character and hence, an average long-term water content of 25 ± 5 % was found more appropriate for these samples. For comparable sedimentary facies, values between 20 and 30 % were used (Anselmetti *et al.,* 2010; Dehnert *et al.,* 2012; Lowick *et al.,* 2015; Buechi *et al.,* 2017). However, the assessment of an appropriate long-term water content is complex but important to avoid age over- or underestimations. It is well known that moisture has an attenuation effect of beta and gamma radiation

(Zimmermann, 1971) which may lead to drastic changes in $D_{total}$ and thereby age determination (Nathan and Mauz, 2008). Between 34 and 49 % of water, in relation to the samples dry weights, were absorbed over a 24 h period by unconsolidated sample material in this study. The highest amounts of water were absorbed by RIN5 and RIN6 which also presented the highest field water contents. For the purpose of comparison, an absolute water content maximum can be set to 50 % which is similar to the highest observed absorption capacity for unconsolidated sample material (Table 2). The $D_{total}$ values are calculated for

water contents between 10 and 50 %, in 10 % steps. In addition, a step with 25 % water content, as assumed appropriate for age determination of RIN5 and RIN6, is included in the comparison. Re-calculated ages (Fig. 13 A. and B.) show a difference



in ages for the different water contents are up to 40 % for Q, 15 to 25 % for F and 40 % for PM and fQ. A water content for Q
and F (RIN1 to RIN4) of 10 % results in statistically consistent ages at 2 σ with fQ and PM (RIN5, RIN6) at 50 %. However,
a 10 % moisture for the coarse-grained samples is unlikely as under current conditions ground water penetrates these layers
and oxidation features are present. Also, 50 % of moisture for fQ and PM are at the limit of maximum water absorption capacity
for unconsolidated material of these samples. It is unlikely that for the consolidated deposits, as found in nature, the maximum
absorption capacity will be equally high and that these samples were saturated to a maximum for over ca. 150 ka. Therefore,
neither 50 % nor 10 % water content are considered representative as a long-term average. Consequently, assumptions of the
water content may well have an effect onto the age discrepancy between RIN5 and RIN6 and the coarse-grained samples from
above (RIN1 to RIN4) but they are not the sole cause for this offset.

### 4.2.2 Alpha efficiency values and age determination

A further cause for the observed discrepancy may be found in the chosen alpha efficiency values (a-values) that account for
the efficiency of alpha particles to produce luminescence which are grain size dependent (cf. Mauz et al, 2006). For the coarse
Q fraction, the need to consider alpha radiation is circumvented by HF etching the outer rind of the grains and thereby removing
the sphere penetrated by alpha particles. However, fQ grains cannot be etched due to the possibility of total dissolution and
therefore an appropriate a-value has to be chosen. In this study, an a-value of $0.05 \pm 0.01$ was used for fQ following Buechi *et
al.* (2017). In the literature, a-values between $0.02 \pm 0.01$ and $0.05 \pm 0.01$ have been presented for fine-grained quartz from
several continents (e.g. Rees-Jones, 1995; Mauz *et al.*, 2006; Lai *et al.*, 2008) and a-values of $0.03 \pm 0.02$ to $0.05 \pm 0.01$ were
used in studies of northern Switzerland (e.g. Gaar and Preusser, 2012; Lowick *et al.*, 2015; Buechi *et al.*, 2017). For the
polymineral fractions, a-values of up to $0.10 \pm 0.01$ and $0.11 \pm 0.01$ (e.g. Rees-Jones, 1995; Lang *et al.*, 2003, Schmidt *et al.*,
2018a) are present in the literature while a-values of between $0.05 \pm 0.01$ and $0.07 \pm 0.02$ (Preusser, 1999; Preusser *et al.*,
2001; Gaar and Preusser, 2012; Gaar *et al.*, 2013; Lowick *et al.*, 2015) are commonly used for polymineral and coarse-grained
feldspar in the studied region. Here, an a-value of $0.05 \pm 0.01$ is used.
The impact of a-value estimates onto the age calculation, can be observed for ages of the investigated samples when re-
calculated for different a-values (0.01 to 0.05, Fig. 13 C. and D.). For fQ and PM, a reduction of the a-value leads to a 13 to
16 % age increase, while for corrected F the a-value has an insignificant impact onto the finite ages (~2 ka). Re-calculated
ages are consistent with each other at 2 σ over the a-value spectrum. Between corrected F and PM samples, lower a-values
(e.g. 0.01) will decrease the age differences. However, the chosen a-value of 0.05 is at the lower limit of values proposed in
the literature. A sample specific determination of a-values is needed to justify the use of a lower a-value. However, in absence
of sample-specific a-values and limited gain of age precision, the initially chosen a-value will be retained.
Also, with the lowest possible a-value of 0.01 for fQ, an offset to the Q ages of the top four samples and RIN5 and RIN6 is
with ca. 20 ka still apparent. A further increase of the used a-values will amplify the age offset. Consequently, neither the
chosen a-value nor water content are the sole causes for the observed age offset.




### 4.2.3 Quartz grain-size dependency

A further component, capable of causing discrepancies in ages is the effect of grain-size dependent luminescence characteristics. Here, two different grain size ranges (200-250 and 4-11 µm) were used for age determination and the offset is seemingly marked by the two grain size groups.

A comparison of fine- and coarse-grained quartz of waterlain sediments from the studied region reported ages in agreement for both fractions or older ages for the fine-grained fractions (Lowick *et al.*, 2015). Similarly, in several studies older ages are presented for the fine-grained fractions of alluvial samples, even though, bleaching conditions were expected to be preferential for the fine fractions (e.g. Olley *et al.*, 1998; Colls *et al.*, 2001). Nevertheless, a pattern has been observed for numerous sites from around the globe where fine-grained quartz ages of ~40 ka underestimate ages derived using coarse-grained quartz and/or

independent age controls (cf. Timar-Gabor *et al.*, 2017). A dose dependency (>100 Gy) was linked to this phenomenon and thermal instability was rejected as a potential cause, though, a finite explanation is still lacking. Given the internal consistency of Q ages for RIN1 to RIN4 and findings summarised by Timar-Gabor *et al.* (2017), fQ ages of RIN5 and RIN6 should be regarded as minimum age estimates. This is also applicable for fQ of RIN8 and RIN13.

**5 Conclusions**

The investigated luminescence properties of coarse- and fine-grained quartz, feldspar polymineral fractions of eight samples from a palaeovalley in northern Switzerland were assessed and six were found appropriate for finite dating. For the chosen dose rates (a-value of 0.05, water content of 20 to 25 %), derived age estimates are in good agreement with each other and the chrono-stratigraphic context (Fig. 2). For the upper four samples (RIN1 to RIN4), Q and 'Kars' fading corrected F ages are in

good agreement with each other and with fading corrected PM ages of two samples directly underneath (RIN5, RIN6). Minimum fQ for the two latter are no contradiction. This indicates a deposition of at least 16.6 m during MIS 6 with a rapid transition from lacustrine to colluvial-dominated environments. For the two lowest samples, F and PM measurements exceeded the dating limit at $2*D_0$ but minimum fQ and Q ages indicate that these samples are of similar age or older than the top units.

**Author contributions**

Manuscript conceptualisation and experiment design was conducted by DM with support by FP. DM obtained and analysed the presented data and prepared the manuscript with contributions from all co-authors. Sample context and material was provided by MB and LG. LG contributed Fig. 1 and parts of Fig. 2 while GD was instrumental for funding acquisition.



## Competing interests

The authors declare that they have no conflict of interest.

## Data availability

Data is available upon request.

## Acknowledgments

The authors like to thank Lisa Ahlers for her support in the Freiburg OSL laboratory.

## Financial support

This research was funded through the National Cooperative for the Disposal of Radioactive Waste (NAGRA).

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

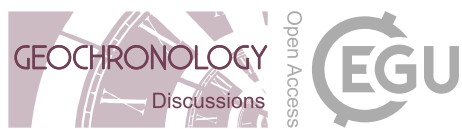

**Figures and Tables**

**Fig. 1** Overview of the study area and location of drill site.

**Fig. 2** Core log and age-depth model of the investigated site. Ages for all minerals and fractions are presented with 1 σ uncertainty. Fading corrected ages are calculated following Kars et al. (2008).

**Fig. 3** Preheat plateau, dose recovery and thermal transfer tests for Q and F of RIN2 and fQ and PM of RIN5.

**Fig. 4** To determine thermal quenching, repeated short shine measurements of Lx/Tx for a given dose of 90 Gy after preheats at different temperatures were conducted on one aliquot of RIN2.

**Fig. 5** Short shine pulse annealing experiment on naturally and laboratory induced doses of two aliquots from RIN2. The IRSL signal is normalised to a test dose and re-normalised to the first measurement.

**Fig. 6** $D_e$ determination of the natural signal using different stimulation temperatures for each three aliquots of RIN2 F.

**Fig. 7** Decay and $D_e(t)$-plots of the natural doses and dose recovery tests for Q and F of RIN2 and fQ and PM of RIN5.

**Fig. 8** Component deconvolution of the natural and regenerative OSL response of RIN2 Q and RIN5 fQ.

**Fig. 9** Extended dose response curves for RIN2 Q and F.

**Fig. 10** Dose recovery test (RIN2 F, 130 Gy given dose) at different preheat temperatures with and without a mounted cardboard barrier to reduce photon passage.

**Fig. 11** Coloured EMCCD image obtained from 100 F grains of RIN2. The grains were placed on a Risø single grain disc with a 10*10 grid of holes. **A.** Holes containing F grains emitting luminesce are presented in green while those without emission are shown in white. **B.** The 3D surface plot emphasises that some grains inherit very bright signals while others are rather dim.

**Fig. 12** $D_e$ distributions of all measured samples and minerals. CAM $D_e$ values are given with 1 σ uncertainty. F and PM $D_e$ values are uncorrected.



**Fig. 13** Ages of **A.** Q and fQ as well as **B.** F and PM (fading corrected after Kars et al., 2008) are shown for water contents
between 10 and 50% and plotted against depth (line symbols). Ages of **C.** fQ as well as **D.** F and PM (fading corrected after
Kars et al., 2008) are shown for alpha efficiency values between 0.01 and 0.05 and plotted against depth (line symbols). No
alpha component was considered for age determination of HF etched Q, but accepted ages are presented for the sake of
completeness. Accepted ages are presented with 1 σ uncertainties as point symbols.


**Table 1**. De values and derived ages for all samples, 1 σ uncertainties are given. The number of accepted/measured aliquots
is provided by n. Ages to be considered for chrono-stratigraphic interpretation are bold.

**Table 2.** Dosimetric data and total dose rates as used for age determination (all values are given with 1 σ uncertainties).



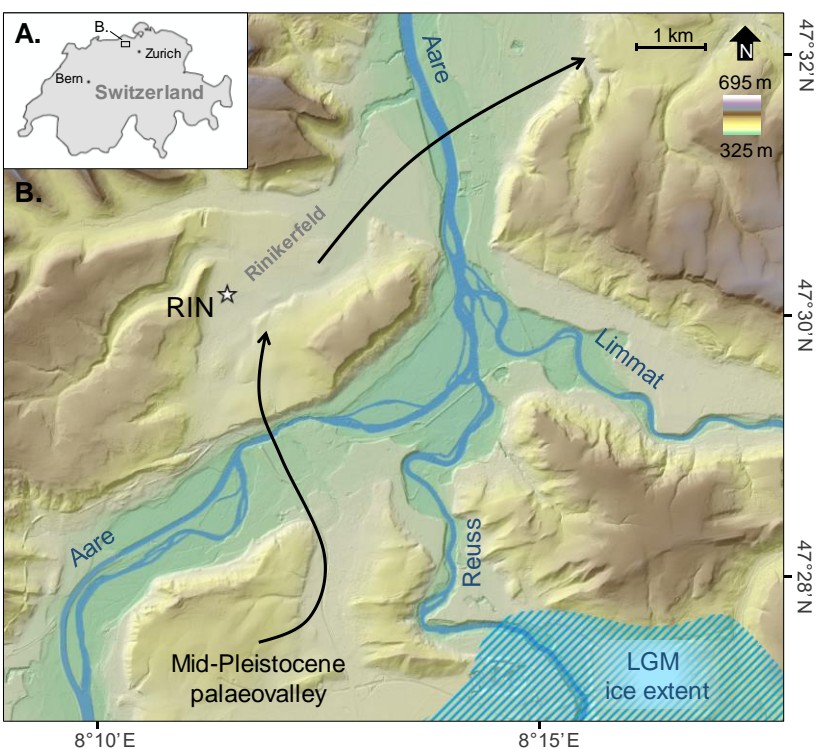


**Figure 1 Overview of the study area and location of drill site (data source: Swisstopo, 2013).**



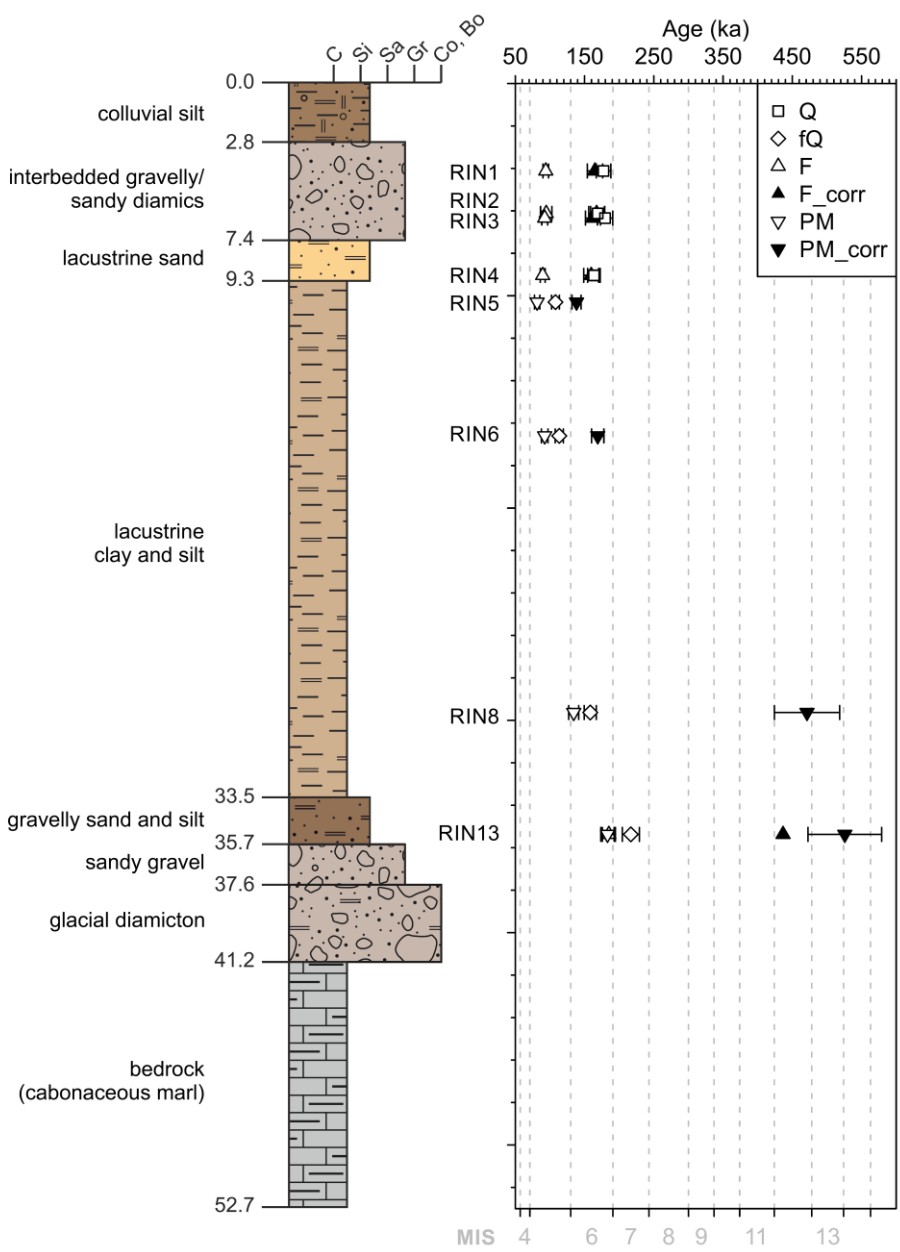

**Fig. 2 Core log and age-depth model of the investigated site. Ages for all minerals and fractions are presented with 1 σ uncertainty.**
**Fading corrected ages are calculated following Kars et al. (2008).**



**Fig. 3 Preheat plateau, dose recovery and thermal transfer tests for Q and F of RIN2 and fQ and PM of RIN5.**




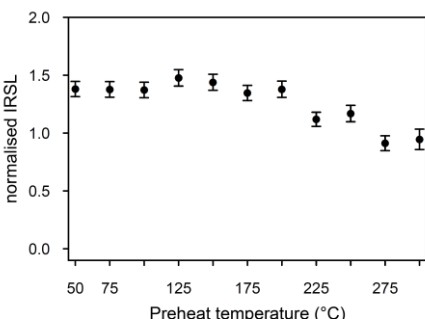

**Fig. 4 To determine thermal quenching, repeated short shine measurements of Lx/Tx for a given dose of 90 Gy after preheats at different temperatures were conducted on one aliquot of RIN2.**

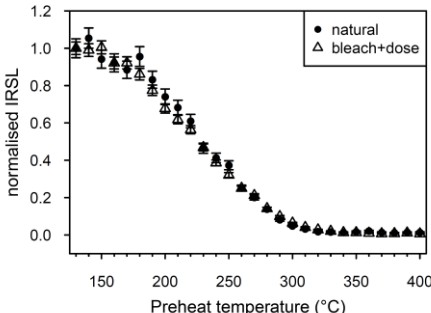

**Fig. 5 Short shine pulse annealing experiment on naturally and laboratory induced doses of two aliquots from RIN2. The IRSL**
**signal is normalised to a test dose and re-normalised to the first measurement.**

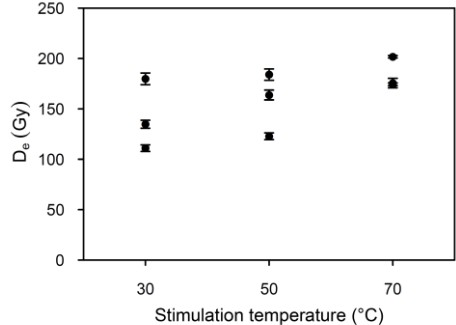

**Fig. 6 $D_e$ determination of the natural signal using different stimulation temperatures for each three aliquots of RIN2 F.**






**Fig. 7 Decay and $D_e$(t)-plots of the natural doses and dose recovery tests for Q and F of RIN2 and fQ and PM of RIN5.**



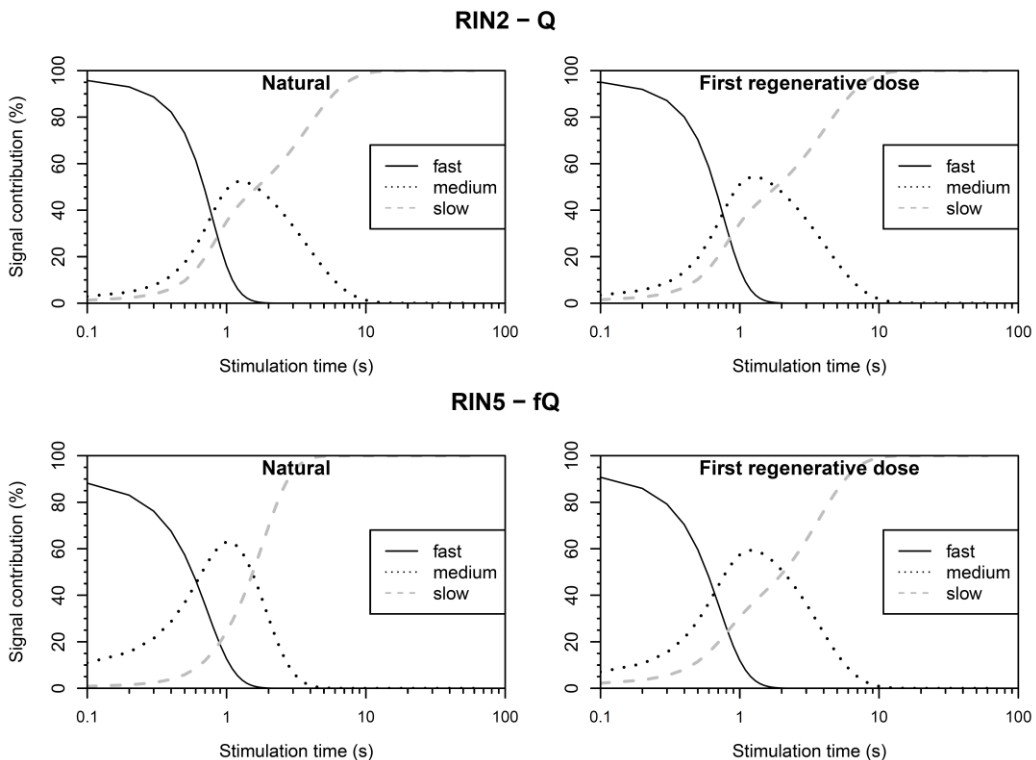

**Fig. 8 Component deconvolution of the natural and regenerative OSL response of RIN2 Q and RIN5 fQ.**




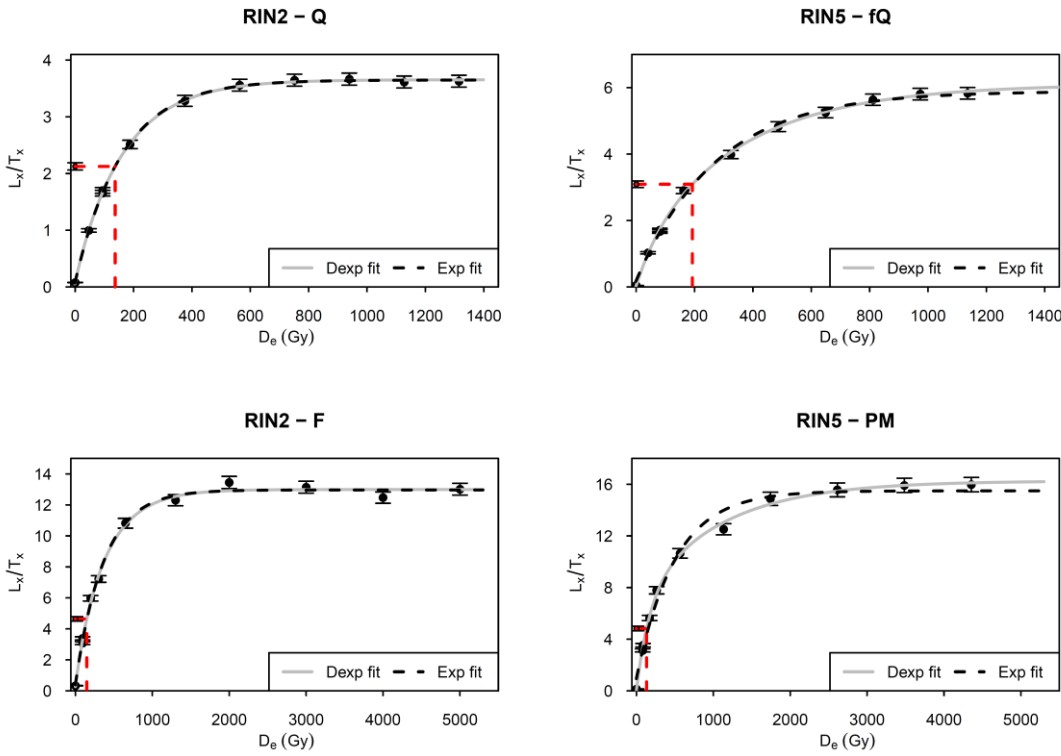


**Fig. 9 Extended dose response curves for RIN2 Q and F.**



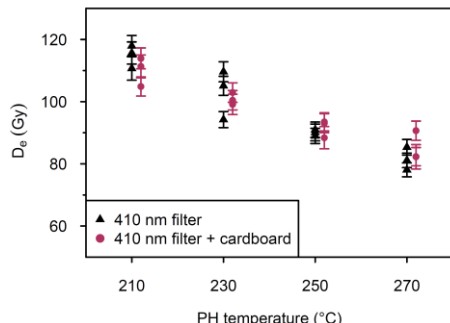

**Fig. 10 Dose recovery test (RIN2 F, 130 Gy given dose) at different preheat temperatures with and without a mounted cardboard barrier to reduce photon passage.**



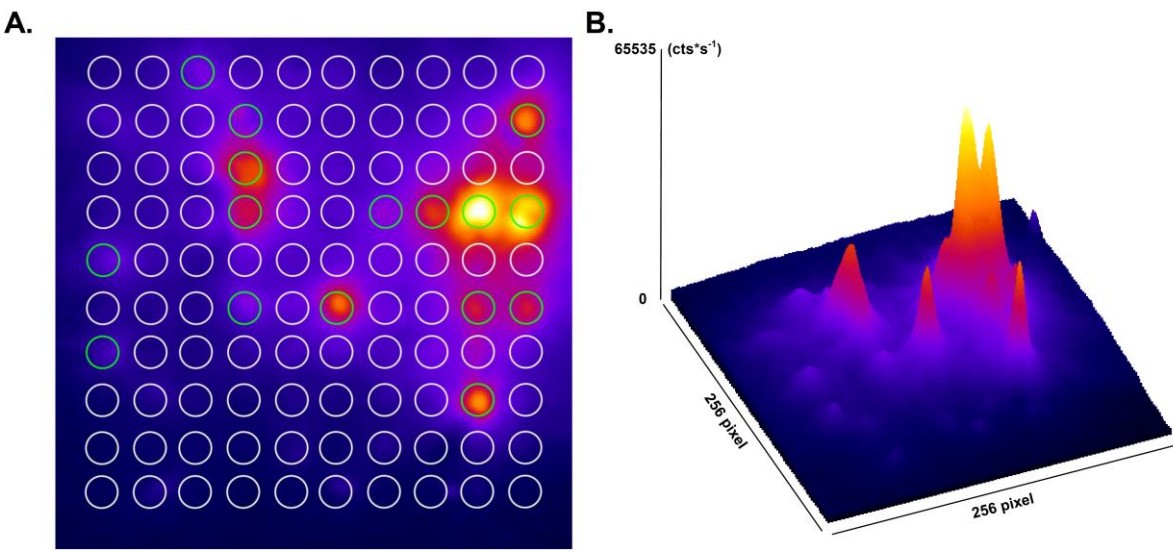


**Fig. 11 Coloured EMCCD image obtained from 100 F grains of RIN2. The grains were placed on a Risø single grain disc with a 10*10 grid of holes. A. Holes containing F grains emitting luminesce are presented in green while those without emission are shown in white. B. The 3D surface plot emphasises that some grains inherit very bright signals while others are rather dim.**








**Fig. 12 D$_e$ distributions of all measured samples and minerals. CAM D$_e$ values are given with 1 σ uncertainty. F and PM D$_e$ values are uncorrected.**





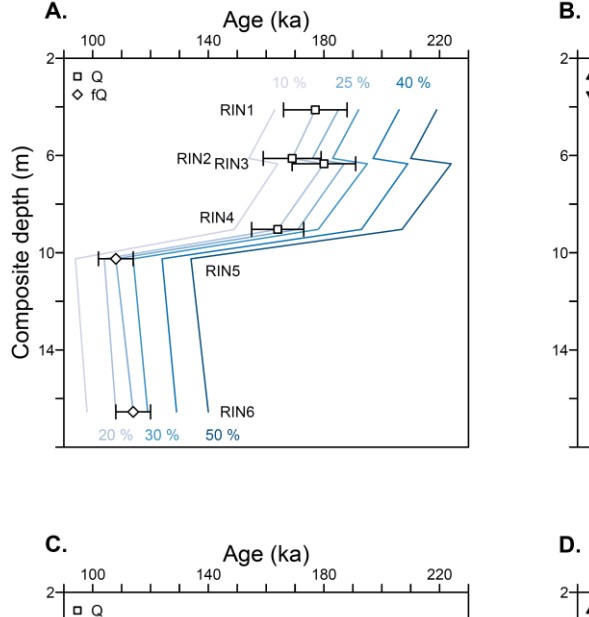

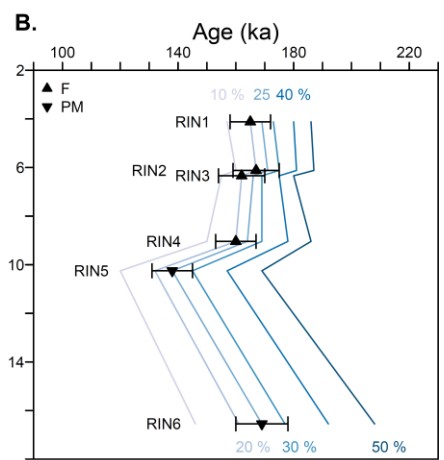

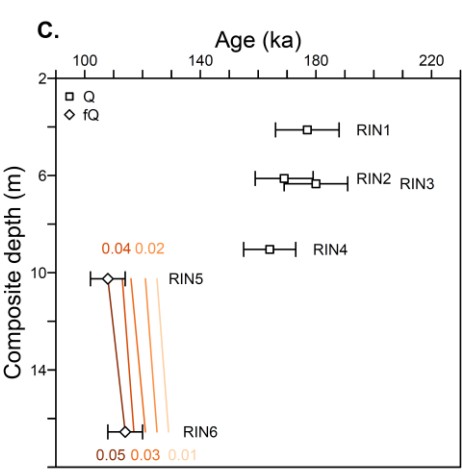

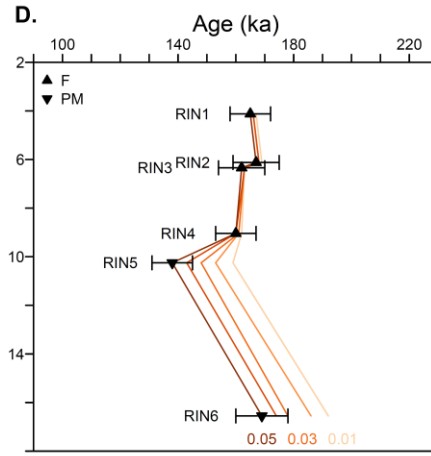

**Fig. 13** Ages of A. Q and fQ as well as B. F and PM (fading corrected after Kars et al., 2008) are shown for water contents between 10 and 50% and plotted against depth (line symbols). Ages of C. fQ as well as D. F and PM (fading corrected after Kars et al., 2008) are shown for alpha efficiency values between 0.01 and 0.05 and plotted against depth (line symbols). No alpha component was considered for age determination of HF etched Q, but accepted ages are presented for the sake of completeness. Accepted ages are presented with 1 σ uncertainties as point symbols.

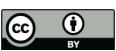

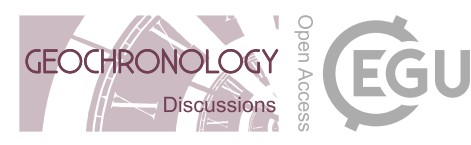

**Table 1. De values and derived ages for all samples, 1 σ uncertainties are given. The number of accepted/measured aliquots is provided by n. Under M, the measured mineral is given. Ages to be considered for chrono-stratigraphic interpretation are bold.**

| Sample code | M | Grain size (µm) | n | OD[A] (%) | 2D0_CAM (Gy) | De_CAM (Gy) | De_Lamothe[B] (Gy) | De_Kars[C] (Gy) | Age_CAM[D] (ka) | Age_Lamothe[B] (ka) | Age_Kars[C] (ka) |
|---|---|---|---|---|---|---|---|---|---|---|---|
| RIN1 | Q | 200-250 | 30/31 | 24 ± 3 | 315.5 ± 16.6 | 149.5 ± 6.9 | - | - | 177.4 ± 10.5 | - | - |
|  | F | 150-200 | 30/30 | 11 ± 2 | 577.2 ± 19.5 | 140.9 ± 3.1 | 214.8 ± 5.4 | 247.2 ± 6.5 | 94.0 ± 4.0 | 143.4 ± 6.3 | 165.0 ± 7.3 |
| RIN2 | Q | 200-250 | 30/31 | 25 ± 3 | 342.5 ± 12.7 | 144.2 ± 6.8 | - | - | 168.9 ± 10.2 | - | - |
|  | F | 200-250 | 30/30 | 14 ± 2 | 626.2 ± 21.7 | 158.4 ± 4.4 | 241.9 ± 7.5 | 280.2 ± 9.4 | 94.6 ± 4.4 | 144.5 ± 7.0 | 167.4 ± 8.3 |
| RIN3 | Q | 200-250 | 30/30 | 25 ± 3 | 317.4 ± 15.4 | 146.3 ± 6.9 | - | - | 179.6 ± 11.0 | - | - |
|  | F | 200-250 | 30/31 | 14 ± 2 | 609.3 ± 18.3 | 151.0 ± 4.0 | 227.7 ± 6.6 | 265.0 ± 8.2 | 92.4 ± 4.2 | 139.3 ± 6.6 | 162.1 ± 7.9 |
| RIN4 | Q | 200-250 | 30/33 | 21 ± 3 | 365.2 ±15.8 | 194.7 ± 7.8 | - | - | 163.5 ± 9.1 | - | - |
|  | F | 150-200 | 30/30 | 6 ± 1 | 640.5 ± 25.6 | 166.8 ± 2.3 | 255.5 ± 4.1 | 297.3 ± 5.5 | 89.5 ± 3.5 | 137.1 ± 5.6 | 159.6 ± 6.6 |
| RIN5 | fQ | 4-11 | 7/7 | - | 430.7 ± 27.7 | 195.0 ± 4.4[F] | - | - | >108.0 ± 5.6[F] | - | - |
|  | PM | 4-11 | 30/30 | 3 ± 1 | 445.2 ± 19.0 | 148.2 ± 1.6 | 208.6 ± 2.4 | 251.6 ± 3.1 | 81.5 ± 3.9 | 114.8 ± 5.5 | 138.4 ± 6.6 |
| RIN6 | fQ | 4-11 | 7/7 | - | 459.0 ± 31.4 | 227.3 ± 5.3[F] | - | - | >113.5 ± 5.9[F] | - | - |
|  | PM | 4-11 | 7/7 | - | 457.5 ± 44.0 | 187.5 ± 3.7 | 271.7 ± 5.9 | 343.0 ± 8.6 | 92.4 ± 4.7 | 133.9 ± 6.9 | 169.0 ± 8.9 |
| RIN8 | fQ | 4-11 | 7/7 | - | 577.1 ± 26.8 | 365.0 ± 10.4[F] | - | - | >158.2 ± 8.8[F] | - | - |
|  | PM | 4-11 | 7/7 | - | 480.1 ± 39.5 | 313.7 ± 7.6 | 559.7 ± 19.3 | 1100.5 ± 96.7 | 134.3 ± 7.2 | 239.6 ± 14.1 | 471.2 ± 47.1 |
| RIN13 | Q | 200-250 | 19/26 | 56 ± 10 | 274.1 ± 16.3 | 255.4 ± 34.2[G] | - | - | >182.8 ± 25.6[FG] | - | - |
|  | fQ | 4-11 | 7/7 | - | 545.2 ± 28.4 | 388.0 ± 12.0[F] | - | - | 216.8 ± 12.3[G] | - | - |
|  | F | 200-250 | 30/30 | 17 ± 3 | 650.5 ± 31.8 | 412.8 ± 14.1 | >694.5[E] | >973.7[E] | 185.1 ± 9.5 | >311.4[E] | >436.6[E] |
|  | PM | 4-11 | 7/7 | - | 514.5 ± 38.6 | 332.7 ± 8.2 | 542.4 ± 17.3 | 955.4 ± 85.6 | 183.1 ± 9.7 | 298.5 ± 17.0 | 525.8 ± 53.2 |

[A] Overdispersion was calculated using the CAM (Galbraith et al., 1999).

[B] A g-value (normalised to 2 days) of 3.1 ± 0.6 was used for F of RIN1 to RIN4 and RIN13 and of 2.4 ± 0.6 for PM of RIN5 to RIN13.

[C] A ρ'-value of 2.21*10-6 was used for F of RIN1 to RIN4 and RIN13 and of 1.89*10-6 for PM of RIN5 to RIN13.

[D] Ages were calculated using the weighted mean of the uncorrected De values derived with the CAM (Galbraith et al., 1999).

[E] Out of 30 measured aliquots, 12 (Lamothe) and 19 (Kars) aliquots are in saturation following fading correction and, therefore, the values shown are derived using a CAM on a truncated distribution.

[F] Regarded to be minimum estimates due to grain size dependent De underestimation.

[G] Derived CAM is based on a truncated distribution and results should be considered as minimums.



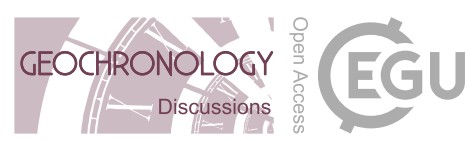

**Table 2. Dosimetric data and total dose rates as used for age determination (all values are given with 1 σ uncertainties).**

| Sample code | Core depth (m) | WC field^A (%) | WC max^A (%) | WC used^A (%) | Radionuclide concentrations U (ppm) | Th (ppm) | K (%) | $D_{cosmic}$ (Gy ka$^{-1}$) | $D_{total}$ Q and fQ^B (Gy ka$^{-1}$) | $D_{total}$ F and PM^C (Gy ka$^{-1}$) |
|---|---|---|---|---|---|---|---|---|---|---|
| RIN1 | 4.12 | 16 | 39 | 20 ± 5 | 0.81 ± 0.17 | 3.52 ± 0.24 | 0.44 ± 0.03 | 0.135 ± 0.014 | 0.84 ± 0.05 | 1.50 ± 0.06 |
| RIN2 | 6.12 | 17 | 46 | 20 ± 5 | 0.69 ± 0.15 | 3.31 ± 0.22 | 0.53 ± 0.04 | 0.108 ± 0.011 | 0.85 ± 0.05 | 1.67 ± 0.08 |
| RIN3 | 6.34 | 19 | 34 | 20 ± 5 | 0.93 ± 0.18 | 3.20 ± 0.22 | 0.49 ± 0.04 | 0.105 ± 0.011 | 0.81 ± 0.05 | 1.63 ± 0.08 |
| RIN4 | 9.04 | 18 | 40 | 20 ± 5 | 1.44 ± 0.26 | 4.90 ± 0.30 | 0.77 ± 0.05 | 0.081 ± 0.008 | 1.19 ± 0.07 | 1.86 ± 0.07 |
| RIN5 | 10.25 | 24 | 48 | 25 ± 5 | 1.54 ± 0.27 | 6.00 ± 0.40 | 0.96 ± 0.07 | 0.073 ± 0.007 | 1.79 ± 0.09 | 1.82 ± 0.09 |
| RIN6 | 16.55 | 23 | 49 | 25 ± 5 | 1.75 ± 0.29 | 6.70 ± 0.40 | 1.18 ± 0.08 | 0.044 ± 0.004 | 2.00 ± 0.10 | 2.03 ± 0.10 |
| RIN8 | 29.63 | 19 | 42 | 20 ± 5 | 2.26 ± 0.35 | 6.80 ± 0.50 | 1.28 ± 0.09 | 0.020 ± 0.002 | 2.31 ± 0.13 | 2.34 ± 0.13 |
| RIN13 | 35.37 | 15 | 45 | 20 ± 5 | 1.53 ± 0.24 | 5.50 ± 0.40 | 0.99 ± 0.06 | 0.015 ± 0.002 | 1.40 ± 0.06 (Q) / 1.79 ± 0.10 (fQ) | 2.23 ± 0.11 (F) / 1.91 ± 0.10 (PM) |

A Water content as measured from the samples (field), maximum absorption capacity as measured in laboratory tests (max) and as used for De determination (used).

B Alpha efficiency of 0.05 ± 0.01 was assumed for fQ.

C Alpha efficiency of 0.05 ± 0.01 and potassium content of 12.5 ± 0.5 % were assumed.

725