# Peer review of "Luminescence properties and dating of glacial to periglacial sediments from northern Switzerland"

_Geochronology, 2020_

## Referee Comment (RC1) · Anonymous Referee #1 · 20 Jul 2020

**Referee comment:**
**Luminescence properties and dating of proglacial sediments from northern Switzerland**

**General comments**

This manuscript presents new data based on luminescence analyses of minerogenic sediments in a palaeovalley in Switzerland. As the title indicates, most focus is on the dating and quite little is spent on the implications of the actual ages and their geological context. The luminescence analyses are thorough, though: the authors have used quartz, feldspar and polymineral fractions of several different grain sizes and carried out different tests to evaluate the luminescence properties of the sampled sediment. Both dose and dose-rate related issues are discussed. The discussions are relevant and interesting for luminescence users not only in the Alps, but also in other parts of the world.

Apart for some specific comments, generally about clarifications, and some minor technical corrections as listed below, my main objection or concern about this manuscript is its structure. Though the headings follow the normal IMRAD standard, content-wise there is a mix between Methods, Results and Discussion. When reading Results, in particular, it is like following the project and measurements as they developed. Results are presented, comparisons and references to other studies are made to evaluate data and motivate the next methodological step, which is then described, etc. In a way, this is quite nice, and likely saves some flipping back and forth between pages to check Methods for what was done and how compared to the Results etc, but it also means that, for example, some details regarding methods are presented first in Results, making it hard to find information, and it is in places not that easy to distinguish what are the new results of the authors' in the text (though obvious from tables and figures).

1. Does the paper address relevant scientific questions within the scope of GChron? Yes
2. Does the paper present novel concepts, ideas, tools, or data? Yes, new data from a combination of existing methods and ideas
3. Are substantial conclusions reached? Yes
4. Are the scientific methods and assumptions valid and clearly outlined? Yes, valid and largely clearly outlined
5. Are the results sufficient to support the interpretations and conclusions? Yes
6. Is the description of experiments and calculations sufficiently complete and precise to allow their reproduction by fellow scientists (traceability of results)? Overall, yes. Some minor details could be clarified.
7. Do the authors give proper credit to related work and clearly indicate their own new/original contribution? Proper credit, yes, own results partly and slightly lost in text.
8. Does the title clearly reflect the contents of the paper? Yes, but see specific comment about proglacial below.
9. Does the abstract provide a concise and complete summary? Yes

10. Is the overall presentation well structured and clear? Yes and no, see above for general comment.

11. Is the language fluent and precise? Yes, largely. Some spelling or grammar mistakes, but with one or two exceptions nothing that hampers understanding.

12. Are mathematical formulae, symbols, abbreviations, and units correctly defined and used? Yes, overall

13. Should any parts of the paper (text, formulae, figures, tables) be clarified, reduced, combined, or eliminated? Yes. Fig. 12 needs revision (too small and/or light text). Regarding text, see general comment above.

14. Are the number and quality of references appropriate? Yes

15. Is the amount and quality of supplementary material appropriate? There is none.

**Specific comments**

- L83-86: Please provide some more detail about the sediments including interpretation. Are the sediments that you date really proglacial like the title says? It is not mentioned here. The dated sediments are only described as "sandy gravel" (no glacier indicated), "lacustrine" (no glacier indicated) and "diamicts and colluvium" (no glacier indicated).

- L125: Please explain the combined recycling and IR depletion step. Do you mean that this is one set of measurement (one column (or row) in the sequence) only or is it two? Combined to me sounds as if it is one, but then you would either have a proper recycling (using blue stimulation only as after the first regen doses) or an IR depletion test (using infrared + blue stimulation), not both. And if you have two measurements, it is not combined.

- L146 (and thereabouts): What was the cutheat?

- L157: Against what value was the normalisation done?

- L184/214: What do you mean by "rarely inherit luminescent behaviour"? That the sediments are efficiently bleached (no inherited signal/dose)? That the grains do not reflect the properties of the source rocks? Please explain.

- L187: The reduction of the signal to background level after 15-20 s is not shown in the figure 7, which is referred to. The Q and fQ plots there only cover 6 s. Also, the background seems to me to be reached already after 2-4 s.

- You touch upon the issue of incomplete bleaching in a couple of places, but you do not really evaluate/discuss it for your samples, apart from a few lines about RIN 8 and 13 (L300-). Yet, also samples RIN1-4 have significantly skewed dose distributions, and diamictons (RIN1-3) are not obviously well-bleached sediments. With easy-to-bleach quartz ages older than or similar to harder-to-bleach feldspar ages, it may not be a problem, but that is also an interesting result that could be discussed, even if only briefly.

- Will the implications of the ages be presented and discussed in some other paper? If not, I think that part should be expanded a bit here.

**Technical corrections**

- L30: replace "waters which both" with "waters, both of which"
- L80: remove comma after Rinikerfeld
- L85: "transition" should be "transitions"
- L87: remove "in" after "10 cm"

- L89: exchange "embrace" for a verb that more clearly indicates over- and underlying, rather than incorporating
- L118: "an" should be "a"
- L184: Please add something after "1%". Is it 1% of the aliquots, of the grains…?
- L184: replace "BSL" with blue light
- L186: clarify that 200 grains are for Q only, not fQ
- L186: "rains" should be "grains"
- L208: "FQ" should be "fQ"
- L215/219: Please check figure order. Fig. 10 must be referred to before Fig. 11.
- L241: Add "Of" in front of "the measured"
- L241: Replace "saturation only allowing" with "saturation, allowing only"
- L253: remove "RIN13 with" before "216.8"
- L258 and elsewhere: It is confusing that you use the same letter symbol, D, both for dose and dose rate. A subscript (here $_e$ and $_{total}$) typically indicates different versions of the same entity, not different entities. When I first read "$D_{total}$" I interpreted it as 'total dose', which did not make sense, and it was only after looking at table 2 that I found out that you meant 'total dose rate'. Please exchange one of the symbols.
- L266: Please rephrase the last sentence. I do not understand what you mean. What is the offset less pronounced than? The difference between the Q and fQ? The offset between F (or PM) and Q (or fQ)?
- L280 and elsewhere: Either use ' ' for both corrections or for none. Ie, either "Lamothe correction" and "Kars correction", or "'Lamothe' correction" and "'Kars' correction".
- L296: remove "according"
- L297: replace "this" with "the age"
- L311: specify which g-value is referred to (F or PM)
- L317: Should it be "1.9 % per decade" or do you mean that the values are 0.5-1.9 % lower than some other value? Please clarify.
- L328: Add "the" after "However, "
- L347: Replace "were" with "have been"
- Fig. 1: check spelling of diamicts and carbonaceous
- Fig. 9 caption: add also RIN 5 fQ and PM.
- Fig. 11 caption: check spelling of luminescence on second line
- Fig. 11: the greens rings are hardly discernible from the white rings, consider using another colour
- Fig. 12: the text inside the plots is not legible, it is way too small. The light grey colour in the four lower plots is hard to see, particularly for the text.
- Fig. 13. Would it be possible to indicate the stratigraphy (e.g. unit numbers and boundaries) in these plots? It would help a reader to remember the stratigraphic context and which samples belong where.

---

## Referee Comment (RC2) · Sebastian Kreutzer (Referee) · 21 Jul 2020

**Contribution summary**

The manuscript presents a dating study from a palaeovalley (Rinikerfeld) from northern Switzerland. The authors retrieved eight samples from a drilling campaign for luminescence dating. Prepared were either the quartz, feldspar or the polymineral fraction using three different grain-sizes. The preferred grain-size fraction was altered with the sedimentological environment. Along with the chronology, the study aims at providing better insight into the luminescence characteristic to assess the potential for further luminescence dating studies that region.

[Figure]

**Recommendation**

I suggest that the manuscript changes a little bit the story and becomes transferred to another Copernicus journal, e.g., E & G (https://egqsj.copernicus.org).

**Justification**

The manuscript presents a concise luminescence-dating study for different minerals, including tests, such as preheat/dose recovery tests or thermal transfer tests from a drilling site in the foreland of the Swiss Alps. All tests are reasonably explained, justified, and they help to support the chronological findings.

However, the manuscript claims to target, specifically, luminescence properties of "proglacial sediments from northern Switzerland", but it remains foremost (and nothing is wrong with it!) a luminescence-dating study. The manuscript conclusion by the authors may best reflect this assessment. The performed tests and measurements are not particular, but something luminescence-dating studies present all the time to increase the confidence in the results. The study on eight samples characterises, as a byproduct, to some extent the luminescence properties of those samples. Nevertheless, it does not investigate luminescence properties in general for the area or directly compares findings from a large dataset (e.g., as a meta-study).

Hence, the exciting part of the study is the chronology itself concerning the palaeovalley. This is the story the manuscript should exploit and detail further. Currently, only a few lines (including the conclusion) wrap the setting of the site and its geomorphological and geological background. Thus, it falls short and more context would also help to understand the chronological findings better.

At the end of the introduction the authors wrote that they "asses" luminescence prop-

erties of the samples from the site, but again, it does not evolve beyond standard tests.

Therefore, I suggest that the authors alter the story a little bit towards a geoscientific focus, add some details as requested below (e.g., dose rate) keep what they have and submit the manuscript to, e.g., E & G (https://egqsj.copernicus.org).

I left a couple of comments below. Mainly referring to some glitches here and there, except the missing results and discussion on the dose rate (the water content is discussed though) and the sketchy discussion on the fine grain quartz age underestimation, nothing critical.

**General comments**

1.  The manuscript reads clear, and the preparation of figures and tables is good. Besides, the authors use many abbreviations in the figures that remain unexplained on top of rather short figure captions. The latter is not necessarily bad, but the manuscript may want to address a broader audience. Currently, the apparent target audience is readers with a background in luminescence dating working in the Swiss Alps. However, other readers may want to have a look into the article as well. Hence, figure captions and abbreviations should elaborate a little bit more, and figures should be more self-explanatory.

2.  The manuscript oddly seems to focus a lot on the differences between coarse and fine grain quartz results, towards an interpretation of an age underestimation of the fine-grain quartz fraction compared to the coarse grain fraction. Indeed, such an age underestimation has been reported in the literature, and I do not doubt these findings for the particular sites.

    However, in the presented manuscript, only for one out of eight samples, a comparison of both grain size fraction is presented. Both numerical results overlap

within uncertainties. All other samples report results, either for the coarse grain (top of the composite profile) or for the fine-grain quartz fraction (lower part). The author's statement on a potential age "underestimation" seems to be trigger by an age inversion in the profile. This age inversion is also present for the fading corrected fine-grain polymineral fraction. Why does this suddenly lead to the conclusion that the fine-grain quartz ages are underestimated (in comparison to the quartz coarse grain ages)? I got the impression that the authors had this idea of a fine-grain quartz underestimation in mind and then tried to see this pattern in their data. It is one possibility that deserves to be discussed, weakly supported by the data though.

3. Technical detail: The chosen format to apply units is odd, e.g., the authors wrote "471.2 ± 47.1 and 525.8 ± 53.2 ka" instead of "471.2 ± 47.1 ka and 525.8 ± 53.2 ka". There is a general pattern in the manuscript and it should be corrected throughout. If I understand the author guidelines of *GChron*[1] correctly, it should even read, e.g., "(525.8 ± 53.2) ka" because the linked SI brochure refers to the *Guide to the expression of uncertainty in measurement*; I might be wrong. Besides, I suggest to round values to meaningful digits. If the age uncertainty is around 10% the number after the digit does not tell much.

**Detailed comments**

Abstract

1. Line 17–18: The statement that the fine-grain quartz ages are underestimated compared to the coarse grain ages does not seem to be supported by the data for three reasons: (1) The authors do not systematically compare coarse grain
* * *
[1]https://www.geochronology.net/for_authors/manuscript_preparation.html

and fine grain quartz ages, they do this for sample RIN13 only. (2) For this particular sample, coarse and fine grain quartz ages overlap within $1\sigma$. (3) Table 1 marks the coarse grain age as "underestimated due to grain size-dependent $D_e$ underestimation", not the fine-grain quartz fraction.

It does not mean that the fine-grain ages are per se not underestimated (e.g., compared the general pattern observed and compared to the profile figure with corrected feldspar ages). Still, the abstract should reflect the essential outcome of the study with regard to the data.

2. Line 19–20: That the dating reveals a rapid deposition during the (at least) MIS 6 was first mentioned in the Conclusion. Doubtlessly, it had no particular relevance to the authors, given the overall scope, but the abstract leaves the reader with the impression that this point will be detailed in the manuscript.

Main text

1. Line 49: "Feldspar" → "feldspar"

2. Line 55: Please report $g$-values normalised to two days or report the $t_c$ value otherwise it will be impossible for readers to compare these values with other findings from the literature.

3. Line 63: I am not sure whether Thiel et al. (2011) should be mentioned as well here (for the 290 °C)?

4. Line 73: Remove "Scientific"

5. Line 101: ca $10^6$ should suffice.

6. Line 110: Add references for the calibration quartz

7. Line 115: Better "when comparing", because this is what people do when they double check your protocol parameters.

8. Line 131: "gamma-ray"

9. Lines 131–132: There is something wrong with the citation chain:

For the fine grain quartz fraction Buechi et al. (2017, p. 57) wrote: *"For fine-grained quartz an a-value of 0.04 ± 0.02 has been incorporated to account for the variability of the values reported in literature (Rees-Jones, 1995; Mauz et al., 2006; Lai et al., 2008)."*.

For the polymineral fraction Buechi et al. (2017, p. 57) reported: *"The effect of alpha irradiation was considered with an a-value of 0.05 ± 0.01 for PM fractions (Preusser, 1999b; Preusser et al., 2001)."*.

Where Preusser (1999b) is the here cited Preusser (1999). It was Preusser et al. (2001) who reported $a$-values for the polymineral fraction as quoted in line 132.

Contrary, Preusser (1999) reported four IRSL $a$-values without uncertainty with a mean of 0.05 and a standard deviation of 0.00 (all values show 0.05; their Table 2). More important is that the applied protocol is not similar to what was applied by the authors here.

In Sec. 4.2.2, the authors detail various possibilities and discuss whether the selected $a$-value is justified. My impression is that the here applied $a$-values were used, because they had been always used for samples from that region. This might be justified, but it also shows that it should be re-measured at some point. In either case, the chosen values need the proper reference.

10. Line 130: U, Th, K concentration values were deduced from gamma-ray spec-
    trometry but only summarised values are presented. What about radioactive dis-
    equilibria?

    Your environment is undoubtedly very challenging regarding the dose rate, so
    maybe you can present a few more results regarding the nuclide concentrations?
    For example, as a plot normalised to Th-232 (cf. Guibert et al., 2009), this would
    give a good indication. If this appears to be too much, the authors can copy
    and paste the data from the VKTA into a supplementing document and add one
    sentence to the main text addressing the possibility of radioactive disequilibria.

11. Line 133: Gaar et al. (2013) confirm Huntley and Baril (1997); which is very
    reassuring. However, (1) they report $12.9 \pm 0.4\,\%$ and (2) they argue for the
    application of the 95 % confidence interval for the potassium concentration (citing
    Huntley and Baril 1997), means ca $12.5 \pm 1\,\%$. Since the authors cited both
    references, they should make elaborate why they did not follow the suggestion by
    Gaar et al. (2013).

12. Lines 140–180 (Sec. 3.1 Performance test): It should read "tests" and please add
    further subsections so that the results for the quartz and the feldspar/ polymineral,
    can be more easily separated.

13. Line 164: It is not really a different "preheat behaviour" but a completely different
    design of the heating element and the thermo couple and its feedback electronic.
    So perhaps: "different technical design"?

14. Line 198: "Chinse" → "Chinese"

15. Line 230: The obtained overdispersion value also depends on the initial $\sigma_b$; if set.
    Was it different from zero?

16. Line 237–238: I am not sure whether CAM is the most suitable model. The authors should double-check the findings by Heydari & Guérin (2018). I also suggest adding one or two dose-response curves from the lower part of the profile.

17. Lines 257–258 ("However, if ..."): The $D_e$ is not a good indication because it is a function of the dose rate and should only be used if the dose rate is homogenous over the profile. Besides, the fading corrected feldspar age appears to be also slightly younger (within $2\sigma$ ok) for the Kars model. My point: If the authors want to keep that argumentation, they should extend the description of the environmental setting and the dose rate. Ages should not be disconnected from the sedimentological environment. For example, why did not a "facies change" (maybe it is not) cause all these "problems"?

18. Lines 261: The purple density curve in RIN13 looks somehow skewed.

19. Lines 269–270: Preusser et al. (2014) wrote that they followed Auclair et al. (2003). Perhaps the latter one is the better reference to cite, or at least in combination with the first. Besides, it appears that Preusser et al. (2014) did not normalise their values to $t_c$ as done by Auclair et al. (2003). In that case, the $g$-value will be slightly different than "expected". The authors may want to add a plot showing their fading measurements; then it should become clear. Additionally, Preusser et al. (2014) measured only three points on the time axis. With regard Kadereit et al. (GChron discussion 2020)[2] the obtained $g$-value might be somewhat arbitrary, and so would be the following fading correction.

    Nevertheless, I did not make this a major point for two reasons: (1) The manuscript by Kadereit et al. will likely be rejected and not become published (though the discussion is online and outlines the general problem). (2) Fading measurements and corrections are a tedious business. The approach chosen
* * *
[2]https://gchron.copernicus.org/preprints/gchron-2020-3/

by the authors might be ok; it might be not. Without further age information, in particular, in the lower part of the composite profile, it is impossible to say.

20. Lines 273: Unfortunately, the function the authors applied to corrected the ages after Lamothe et al. (2003) for fading has a (recently discovered) bug (https://github.com/R-Lum/Luminescence/issues/96). The consequence of the bug is that the uncertainty of the fading corrected ages is lower than it should be because the error of the $g$-value goes into the calculation with a weighting that does not seem to be justified. Of course, this is nothing I hold against the authors. I just wanted to mention it here.

21. Line 275: Please mention the $t_c$ value along with your $g$-value, otherwise they are not comparable. Please add throughout the manuscript.

22. Lines 295–298:I was wondering whether the $D_0$ criterion has any substance after the value became corrected for fading after Kars et al. (2008)? With the correction, the $D_e$ is deduced from a new, simulated, dose-response curve. The $D_0$ of the simulated curve should be the new reference, not the faded dose-response curve. Did I overlook something?

23. Lines 335–336: I do not agree that based on these findings, the logical conclusion is that coarse and fine grain quartz ages are different. The profile may have some hiatus going along with the age inversion. The reason for this age inversion is not necessarily grain-size related.

    Do the authors have granulometric data from the core?

24. Lines 340–341: The last point appears like an appendix in this sections and it leads to nothing further. Is this maybe some kind of leftover from a discussion the authors wanted to engage but did not?

[Figure]

25. Lines 343–365: This is a helpful discussion of different scenarios and justified. However, it should engage a more general discussion on dose-rate scenarios (which does not exist yet).

26. Lines 366–389 (4.2.2 Alpha efficiency values and age determination): This sub-section renders a potentially fascinating discussion. The problem I have with this section, in particular the first part, is that it does not read clearly but mixes different aspects. For example, after reading the section, my conclusion was that the chosen $a$-value of $0.05 \pm 0.01$ is the less justified value. The reasoning is that somehow all goes back to Preusser (1999) and Preusser et al. (2001) Although for Preusser et al., 2001 I am not sure whether it does not resembles values from Preusser, 1999 (?).

    Means, in the worst case, the selection bases on four values with rather low $a$-values, e.g., for the polymineral and feldspar fraction. By contrast, the majority of the other articles would favour higher values. Besides, Schmidt et al. (2018) presented an extended dataset of $a$-values (IRSL and pIRIR$_{290}$), though the focus was pIRIR$_{290}$, which was not measured here.

    Of course, it does not mean that the value is wrong, but the arguments presented by the authors indicate that (as even alluded in the manuscript) that they should remeasure the value.

27. Line 373–374: Please correct the reference or the $a$-value (see above)

28. Line 391: Something is missing in the section title. Perhaps: "Quartz age grain-size dependency" or "Grain-size dependency of the quartz ages"

29. Lines 391–403 (Sec. 4.2.3): The section is very brief and, in my opinion, does not add to the understanding of the "age discrepancies" (if there is any, see comments above) and it does not discuss the quartz grain-size dependency as announced in the section title. Instead, it provides a brief, selective review of other

findings, and it concludes that the lowermost two samples should be regarded as minimum ages. I would support the conclusion, but not the reasoning.

30. Line 400: Timar-Gabor et al. (2017) wrote:

   *On the other hand, the age discrepancy of SAR-OSL ages previously reported for Romanian and Serbian loess for ages beyond ∼40 ka (equivalent doses >∼100 Gy) was also found to be characteristic of Chinese loess. It is thus believed that this is potentially a global phenomenon, affecting previously-obtained chronologies worldwide, and further increasing concerns for the accuracy of silt-sized SAR-OSL ages in this high dose range.* (Timar-Gabor et al., 2017, p. 470).

   Timar-Gabor et al. (2017) expressed a guess or hypothesis as part of the conclusion. This conclusion, however, should not be become some statement on the "pattern around the globe'. At least the cited study does not provide the data to it.

   Furthermore, Timar-Gabor et al. (2017) refer, first of all, to own observations from Romanian and Serbian loess comparing 4–11 $\mu$m (fine grain) and 63–90 $\mu$m (their coarse grain). This is not similar to what is presented in the manuscript for *GChron*.

31. Lines 411: The conclusion should reflect the results and discussion of the manuscript. The depositional history was not discussed in the manuscript and came here by surprise.

Figures and tables

1. Figure 1

   - Do the authors have other ice extent data to show? Perhaps the LGM ice extent is a nice to have, but of limited relevance given the age results.
   - "A." and "B." is part of the figure, consequently those lettering should be part of the figure caption.

2. Figure 2

   - An y-axis unit (core depth) is missing.
   - I would be good to have some photos showing the core log to better understand the setting. Also, the authors may want to indicate where one core ends and the next starts.
   - What does the upper x-axis ("C, Si, Sa, Gr, Co, Bo" on top the core log) labels? Probably it is obvious, but it is not to me and maybe other readers are not familiar with it as well.
   - The age comparison should be based on 95 % confidence intervals. However, I guess the graph will not scale very nicely given the two lower polymineral ages. This can be fixed by using a non-continuous scale.

3. Figure 3

   - The figure would benefit from more details in the figure caption. Readers not familiar with luminescence dating may struggle to understand the figures. For example: "M/G" probably means measured to given dose, "PH" means preheat etc.

- The inset legend in all the figures in the right column is unnecessary because only one type of data is shown in all figures. It would suffice if the figure title (or subtitle) says "thermal transfer test".

- It is not clear what the data points are displaying. A single measurement with uncertainties? An average with the error bars showing the standard deviation (of the mean)?

4. Figure 4

- The mineral fraction is missing in the figure caption

5. Figure 5

- Same as above, the mineral fraction is missing in the figure caption

6. Figure 6

- Same problem as Fig. 3. The figure captions should explain used abbreviations (e.g., "DR")

- The solid line the curve is not really showing the "given dose decay" but the "luminescence signal decay" or shine-down curve of the natural signal. It is a proxy for the "given dose decay", but is not a "dose decay".

- "Natural decay" might lead to a wrong understanding by others who work with dating methods relying on the "radioactive decay" of isotopes. Perhaps: "natural shine-down curve" or something similar.

- $D_e(t)$ plot was used by Bailey et al. (2003) to identify the partial resetting of the luminescence signal. They shifted and extended the signal integrals slightly at the end. Probably it was not done here, but the figure caption should detailed what was done so that the figure becomes immediately understandable.

- Y-axis labelling should be added to the figure in the right column.

7. Figure 12

- Proper x-axis labelling is missing or figures should align more closely.
- Was a similar bandwidth used for all kernel density curves?
- RIN2: Table 1 reads $158.4 \pm 4.4$ Gy instead of $158.4 \pm 4.3$ Gy in the figure (minor detail, since I argue for meaningful rounding above).

8. Figure 13

- 95 % confidence intervals should be used for the age comparison.
- There is somehow a typo in the figure caption: It reads "Accepted ages are presented with $1\sigma$ uncertainty as point symbol.", however, there is no "point symbol" in the figures.

9. Table 2

- What meant is the internal K-concentration, it should be written.
- "De" should read "$D_e$" (subscript "e")

**Personal note to the authors**

Dear Müller et al.,

I can imagine that you do not agree with my suggestion to transfer the manuscript. To avoid the impression that I am "against" your manuscript, I may add that I sincerely believe that every luminescence-dating study deserves to be published; given that it is free of significant mistakes. Luminescence dating is way too costly to ignore the data

or let them disappear in a drawer. Your study indeed, should be published. However, I think for *GChron* it would need way more tests (e.g., TR-OSL, TL with trap parameters etc.) and a larger dataset. This is nothing I can ask of you. To the contrary, you probably have the perfect for, e.g., E & G with a ready to go chronological part if you extent the geomorphological/geological part.

Nonetheless, to ease the minds and to avoid a heated discussion: If the editor believes that the manuscript is most suitable for *GChron*, I will certainly not further argue against a publication in *GChron*. Moreover, I consider my suggestions as a piece to an open discussion, which is not carved in stone.

**Conflict of interest**

I have no conflict of interest to declare. I am not a beneficiary of the suggested references to be cited. Naturally, the authors are free to reject my reference suggestions.

Sebastian Kreutzer – July 21, 2020

**References**

Auclair, M., Lamothe, M., Huot, S., 2003. Measurement of anomalous fading for feldpsar IRSL using SAR. Radiation Measurements 37, 487–492. doi:10.1016/S1350-4487(03)00018-0

Bailey, R.M., Singarayer, J.S., Ward, S., Stokes, S., 2003. Identification of partial resetting using De as a function of illumination time. Radiation Measurements 37, 511-518. doi:10.1016/S1350-4487(03)00063-5

Huntley, D.J., Baril, M.R., 1997. The K content of the K-feldspars being measured in optical dating or in the thermoluminescence dating. Ancient TL 15, 11–13.

Buechi, M.W., Lowick, S.E., Anselmetti, F.S., 2017. Luminescence dating of glaciolacustrine silt in overdeepened basin fills beyond the last interglacial 37, 1–37. doi:10.1016/j.quageo.2016.09.009

Gaar, D., Lowick, S., Preusser, F., 2013. Performance of different luminescence approaches for the dating of known-age glaciofluvial deposits from northern Switzerland. Geochronometria 41, 65–80. doi:10.2478/s13386-013-0139-0

Guibert, P., Lahaye, C., Bechtel, F., 2009. The importance of U-series disequilibrium of sediments in luminescence dating: A case study at the Roc de Marsal Cave (Dordogne, France). Radiation Measurements 44, 223–231. doi:10.1016/j.radmeas.2009.03.024

Heydari, M., Guérin, G., 2018. OSL signal saturation and dose rate variability: Investigating the behaviour of different statistical models. Radiation Measurements 120, 96–103. doi:10.1016/j.radmeas.2018.05.005

Kadereit, A., Kreutzer, S., Schmidt, C., and DeWitt, R. 2020. A closer look at IRSL

SAR fading data and their implication for luminescence dating, Geochronology Discuss., https://doi.org/10.5194/gchron-2020-3, in review, 2020.

Kars, R.H., Wallinga, J., Cohen, K.M., 2008. A new approach towards anomalous fading correction for feldspar IRSL dating — tests on samples in field saturation. Radiation Measurements 43, 786–790. doi:10.1016/j.radmeas.2008.01.021

Lamothe, M., Auclair, M., Hamzaoui, C., Huot, S., 2003. Towards a prediction of long-term anomalous fadingof feldspar IRSL. Radiation Measurements 37, 493–498.

Preusser, F., 1999. Luminescence dating of fluvial sediments and overbank deposits from Gossau, Switzerland: fine grain dating. Quaternary Science Reviews 18, 217–222. doi:10.1016/S0277-3791(98)00054-7

Preusser, F., Müller, B.U., Schlüchter, C., 2001. Luminescence Dating of Sediments from the Luthern Valley, Central Switzerland, and Implications for the Chronology of the Last Glacial Cycle. Quaternary Research 55, 215–222. doi:10.1006/qres.2000.2208

Schmidt, C., Bösken, J., Kolb, T., 2018. Is there a common alpha-efficiency in polymineral samples measured by various infrared stimulated luminescence protocols? Geochronometria 45, 160–172. doi:10.1515/geochr-2015-0095

Preusser, F., Muru, M., Rosentau, A., 2014. Comparing different post-IR IRSL approaches for the dating of Holocene coastal foresdunes from Ruhnu Island, Estonia. Geochronometria 1–10. doi:10.2478/s13386-013-0169-7

Thiel, C., Buylaert, J.P., Murray, A., Terhorst, B., Hofer, I., Tsukamoto, S., Frechen, M., 2011. Luminescence dating of the Stratzing loess profile (Austria) - Testing the potential of an elevated temperature post-IR IRSL protocol. Quaternary International 234, 23–31.

doi:10.1016/j.quaint.2010.05.018

Timar-Gabor, A., Buylaert, J.P., Guralnik, B., Trandafir-Antohi, O., Constantin, D., Anechitei-Deacu, V., Jain, M., Murray, A.S., Porat, N., Hao, Q., Wintle, A.G., 2017. On the importance of grain size in luminescence dating using quartz. Radiation Measurements 106, 464–471. doi:10.1016/j.radmeas.2017.01.009

---

## Author Response (AR2)

**Response to referee #1 for GChron-2020-15**

We would like to thank anonymous referee #1 for their thorough but positive review as well as their constructive comments that have helped to greatly improve this manuscript. We agree with most comments and have added or adjusted text and figures accordingly. For the cases we did not agree with the recommendations made, we give explanations below (responses in green and italic). Also, we would especially like to thank referee #1 for their spelling and grammar suggestions. This is very much appreciated as none of the authors is a native English speaker.

**General comments**

This manuscript presents new data based on luminescence analyses of minerogenic sediments in a palaeovalley in Switzerland. As the title indicates, most focus is on the dating and quite little is spent on the implications of the actual ages and their geological context. The luminescence analyses are thorough, though: the authors have used quartz, feldspar and polymineral fractions of several different grain sizes and carried out different tests to evaluate the luminescence properties of the sampled sediment. Both dose and dose-rate related issues are discussed. The discussions are relevant and interesting for luminescence users not only in the Alps, but also in other parts of the world.

Apart for some specific comments, generally about clarifications, and some minor technical corrections as listed below, my main objection or concern about this manuscript is its structure. Though the headings follow the normal IMRAD standard, content-wise there is a mix between Methods, Results and Discussion. When reading Results, in particular, it is like following the project and measurements as they developed. Results are presented, comparisons and references to other studies are made to evaluate data and motivate the next methodological step, which is then described, etc. In a way, this is quite nice, and likely saves some flipping back and forth between pages to check Methods for what was done and how compared to the Results etc, but it also means that, for example, some details regarding methods are presented first in Results, making it hard to find information, and it is in places not that easy to distinguish what are the new results of the authors' in the text (though obvious from tables and figures).

It is indeed hard to present this type of data in a meaningful and readily comprehensible way. However, the raised issue about mixing methods, results and discussion is certainly justified. Accordingly, we have adjusted subheadings and re-structure the content where necessary to allow for a more appropriate fit between sub-headings and text. For ease of reading, methods, results and discussion are jointly presented in aspect specific subsections. This was previously done but is now implemented in a more stringent manner. Also, we have emphasised in the text when presented results were from a different study.

1. Does the paper address relevant scientific questions within the scope of GChron? Yes

2. Does the paper present novel concepts, ideas, tools, or data? Yes, new data from a combination of existing methods and ideas

3. Are substantial conclusions reached? Yes

4. Are the scientific methods and assumptions valid and clearly outlined? Yes, valid and largely clearly outlined *See below*.

5. Are the results sufficient to support the interpretations and conclusions? Yes

6. Is the description of experiments and calculations sufficiently complete and precise to allow their reproduction by fellow scientists (traceability of results)? Overall, yes. Some minor details could be clarified. *See below*.

7. Do the authors give proper credit to related work and clearly indicate their own new/original contribution? Proper credit, yes, own results partly and slightly lost in text.

8. Does the title clearly reflect the contents of the paper? Yes, but see specific comment about proglacial below. *See below*.

9. Does the abstract provide a concise and complete summary? Yes

10. Is the overall presentation well structured and clear? Yes and no, see above for general comment. *See below*.

11. Is the language fluent and precise? Yes, largely. Some spelling or grammar mistakes, but with one or two exceptions nothing that hampers understanding. *See below.*

12. Are mathematical formulae, symbols, abbreviations, and units correctly defined and used? Yes, overall

13. Should any parts of the paper (text, formulae, figures, tables) be clarified, reduced, combined, or eliminated? Yes. Fig. 12 needs revision (too small and/or light text). Regarding text, see general comment above. *Fig. 12 has been revised – see below*.

14. Are the number and quality of references appropriate? Yes

15. Is the amount and quality of supplementary material appropriate? There is none.

**Specific comments**

• L83-86: Please provide some more detail about the sediments including interpretation. Are the sediments that you date really proglacial like the title says? It is not mentioned here. The dated sediments are only described as "sandy gravel" (no glacier indicated), "lacustrine" (no glacier indicated) and "diamicts and colluvium" (no glacier indicated).

We agree that the manuscript title might have been misleading and that details about the sediments was to scarce. Indeed, the investigated deposits are of glacial/proglacial to periglacial origin. Therefore, text has been added and the title ('Luminescence properties and dating of glacial to periglacial sediments from northern Switzerland') has been changed accordingly.

• L125: Please explain the combined recycling and IR depletion step. Do you mean that this is one set of measurement (one column (or row) in the sequence) only or is it two? Combined to me sounds as if it is one, but then you would either have a proper recycling (using blue stimulation only as after the first regen doses) or an IR depletion test (using infrared + blue stimulation), not both. And if you have two measurements, it is not combined.

Only an IR depletion test was implemented. Text has been changed accordingly.

**• L146 (and thereabouts): What was the cutheat?**

Only preheats were used as is stated in 2.2 'Sample preparation and measurement': 'All performance tests [and measurements] were conducted using preheating previous to the natural, regenerative and test doses for which the aliquots were heated with 5 °C s-1 to the tested temperature and held for 10 s (Q, fQ) or 60 s (F, PM).' To clarify [...] has been added.

**• L157: Against what value was the normalisation done?**

Normalised was against the same fixed dose of ca. 90 Gy. Text has been added accordingly.

• L184/214: What do you mean by "rarely inherit luminescent behaviour"? That the sediments are efficiently bleached (no inherited signal/dose)? That the grains do not reflect the properties of the source rocks? Please explain.

Meant was that only few of the grains actually give a luminescence signal after artificial irradiation. Text has been changed to clarify.

• L187: The reduction of the signal to background level after 15-20 s is not shown in the figure 7, which is referred to. The Q and fQ plots there only cover 6 s. Also, the background seems to me to be reached already after 2-4 s.

**This is correct and the text has been changed accordingly.**

• You touch upon the issue of incomplete bleaching in a couple of places, but you do not really evaluate/discuss it for your samples, apart from a few lines about RIN 8 and 13 (L300-). Yet, also samples RIN1-4 have significantly skewed dose distributions, and diamictons (RIN1-3) are not obviously well-bleached sediments. With easy-to-bleach quartz ages older than or similar to harder-to-bleach feldspar ages, it may not be a problem, but that is also an interesting result that could be discussed, even if only briefly.

**We agree and have added a text to briefly present these findings.**

**• Will the implications of the ages be presented and discussed in some other paper? If not, I think that part should be expanded a bit here.**

This is a good point and yes there will be a summary paper discussing the sedimentological context and the implications of the derived ages. This manuscript will use ages from outcrops, two cores from the Lower Aare Valley and the core presented here. The PhD student within the project (Lukas Gegg) will present this work.

**Technical corrections**

• L30: replace "waters which both" with "waters, both of which"

**Text has been changed.**

• L80: remove comma after Rinikerfeld

Text has been changed.

• L85: "transition" should be "transitions"

Text has been changed.

• L87: remove "in" after "10 cm"

**Text has been changed.**

• L89: exchange "embrace" for a verb that more clearly indicates over- and underlying, rather than incorporating

Text has been changed.

• L118: "an" should be "a"

Text has been changed.

• L184: Please add something after "1%". Is it 1% of the aliquots, of the grains...?

Text has been changed.

• L184: replace "BSL" with blue light

Text has been changed.

• L186: clarify that 200 grains are for Q only, not fQ

Text has been changed.

• L186: "rains" should be "grains"

Text has been changed.

• L208: "FQ" should be "fQ"

Text has been changed.

• L215/219: Please check figure order. Fig. 10 must be referred to before Fig. 11.

Order has been changed accordingly.

• L241: Add "Of" in front of "the measured"

Text has been changed.

• L241: Replace "saturation only allowing" with "saturation, allowing only"

Text has been changed.

• L253: remove "RIN13 with" before "216.8"

**Text has been changed.**

• L258 and elsewhere: It is confusing that you use the same letter symbol, D, both for dose and dose rate. A subscript (here e and total) typically indicates different versions of the same entity, not different entities. When I first read "Dtotal" I interpreted it as 'total dose', which did not make sense, and it was only after looking at table 2 that I found out that you meant 'total dose rate'. Please exchange one of the symbols.

**Nomenclature has been changed and total dose rates are now presented with DRtotal.**

• L266: Please rephrase the last sentence. I do not understand what you mean. What is the offset less pronounced than? The difference between the Q and fQ? The offset between F (or PM) and Q (or fQ)?

**Following suggestions of referee #2, the entire subsection has been changed and this sentence became redundant.**

• L280 and elsewhere: Either use ' ' for both corrections or for none. le, either "Lamothe correction" and "Kars correction", or "'Lamothe' correction" and "Kars' correction".

The use of ' ' has been adopted and was implemented in a stringent manner.

• L296: remove "according"

Text has been changed.

• L297: replace "this" with "the age"

Text has been changed.

• L311: specify which g-value is referred to (F or PM)

Text has been changed.

• L317: Should it be "1.9 % per decade" or do you mean that the values are 0.5-1.9 % lower than some other value? Please clarify.

Meant was '% per decade' and the text has been changed.

• L328: Add "the" after "However, "

Text has been changed.

• L347: Replace "were" with "have been"

Text has been changed.

• Fig. 1: check spelling of diamicts and carbonaceous

Text has been changed.

• Fig. 9 caption: add also RIN 5 fQ and PM.

Text has been changed.

• Fig. 11 caption: check spelling of luminescence on second line

Text has been changed.

• Fig. 11: the greens rings are hardly discernible from the white rings, consider using another colour

*Lines of the green circles have been dashed to help distinguish between them and the white circles. Unfortunately, other tested colours did not improve distinguishability.*

• Fig. 12: the text inside the plots is not legible, it is way too small. The light grey colour in the four lower plots is hard to see, particularly for the text.

The colour has been changed and the text for RIN13 has been adjusted to make it easier to read.

• Fig. 13. Would it be possible to indicate the stratigraphy (e.g. unit numbers and boundaries) in these plots? It would help a reader to remember the stratigraphic context and which samples belong where.

The top part of the log have been added to the figure.

**Response to referee #2 for GChron-2020-15**

We would like to thank referee #2, Sebastian Kreutzer, for his thorough review as well as his constructive comments that have helped to greatly improve this manuscript. We agree with most comments and have substantially added or adjusted text and figures accordingly. For the cases we did not agree with the recommendations made, we give explanations below (responses in green and italic).

One major point we like to address here is the suggestion to shift the scope of the manuscript towards an environmental study and to transfer it to a different journal. Sebastian argues that we are foremost presenting a luminescence-dating study and that from his perspective substantially more tests and a larger dataset are needed to justify a publication in GChron.

We disagree with this recommendation for the following reasons:

- 1. The scope of this manuscript is geochronology and how to get it 'right'.
  - The study itself is part of a large project investigating the fill of several palaeovalleys within northern Switzerland to ultimately reconstruct the environmental history for the wider region. Implications drawn for the Rinikerfeld are important for and will guide the luminescence dating approach for ten other cores and multiple outcrop samples. For this, the confidence in our results is crucial and, therefore, luminescence aspects have to be examined carefully. As both quartz and feldspar from Switzerland inherit challenging luminescence behaviour, we face a multitude of obstacles/opportunities that cannot be adequately addressed in a manuscript focusing on the reconstruction of past environments. The amount of technical details will defocus any such article and be little to non-accessible for the none-specialist.
- 2. Following the aims and scope on the webpage, GChron is defined as '[..] unified outlet for highquality basic and applied research in geochronology, independent of technique used or timescale considered. Geochronology publishes research in all aspects of geoscience that aim to determine times or rates of geologic events and processes [..]'. Our manuscript presents applied research in geochronology.
- 3. Referee #2 argues that the 'performed tests and measurements are [..] something luminescence-dating studies present all the time'. Indeed the 'standard test suit' is conducted while electron trapping probability assessment, deconvolution of quartz signals, pulse annealing experiments to assure not to measure artificial signal introduced during laboratory irradiation and quality control of reader performance go beyond the extent of most papers that are simply presenting and using luminescence ages for environmental reconstructions.

**Contribution summary**

The manuscript presents a dating study from a palaeovalley (Rinikerfeld) from northern Switzerland. The authors retrieved eight samples from a drilling campaign for luminescence dating. Prepared were either the quartz, feldspar or the polymineral fraction using three different grain-sizes. The preferred grain-size fraction was altered with the sedimentological environment. Along with the chronology, the study aims at providing better insight into the luminescence characteristic to assess the potential for further luminescence dating studies that region.

**Recommendation**

I suggest that the manuscript changes a little bit the story and becomes transferred to another Copernicus journal, e.g., E&G (https://egqsj.copernicus.org).

We regard journals with a focus on the reconstruction of Quaternary environments such as E&G not suitable for the present manuscript.

**Justification**

The manuscript presents a concise luminescence-dating study for different minerals, including tests, such as preheat/dose recovery tests or thermal transfer tests from a drilling site in the foreland of the Swiss Alps. All tests are reasonably explained, justified, and they help to support the chronological findings.

However, the manuscript claims to target, specifically, luminescence properties of "proglacial sediments from northern Switzerland", but it remains foremost (and nothing is wrong with it!) a luminescence-dating study. The manuscript conclusion by the authors may best reflect this assessment. The performed tests and measurements are not particular, but something luminescence-dating studies present all the time to increase the confidence in the results. The study on eight samples characterises, as a byproduct, to some extent the luminescence properties of those samples. Nevertheless, it does not investigate luminescence properties in general for the area or directly compares findings from a large dataset (e.g., as a meta-study).

Hence, the exciting part of the study is the chronology itself concerning the palaeovalley. This is the story the manuscript should exploit and detail further. Currently, only a few lines (including the conclusion) wrap the setting of the site and its geomorphological and geological background. Thus, it falls short and more context would also help to understand the chronological findings better.

At the end of the introduction the authors wrote that they "asses" luminescence properties of the samples from the site, but again, it does not evolve beyond standard tests.

Therefore, I suggest that the authors alter the story a little bit towards a geoscientific focus, add some details as requested below (e.g., dose rate) keep what they have and submit the manuscript to, e.g., E&G (https://egqsj.copernicus.org).

I left a couple of comments below. Mainly referring to some glitches here and there, except the missing results and discussion on the dose rate (the water content is discussed though) and the sketchy discussion on the fine grain quartz age underestimation, nothing critical.

The geoscientific focus will be the subject of a separate manuscript by Gegg et al. that will use this and other cores from the area together with geomorphological and outcrop data. This is done to access the long-term erosion history of the area in context of the siting for the Swiss nuclear waste disposal site. The present contribution represents the rigorous testing program that is used to highlight the potential and limitations of luminescence dating in the region. Adding the results presented in this GChron manuscript would overload the paper with focus of reconstruction of Quaternary environments.

**General comments**

1. The manuscript reads clear, and the preparation of figures and tables is good. Besides, the authors use many abbreviations in the figures that remain unexplained on top of rather short figure captions. The latter is not necessarily bad, but the manuscript may want to address a broader audience. Currently, the apparent target audience is readers with a background in luminescence dating working in the Swiss Alps. However, other readers may want to have a look into the article as well. Hence, figure captions and abbreviations should elaborate a little bit more, and figures should be more self-explanatory.

We agree and figures and figure caption have been adjusted according to the below mentioned specific comments.

2. The manuscript oddly seems to focus a lot on the differences between coarse and fine grain quartz results, towards an interpretation of an age underestimation of the fine-grain quartz fraction compared to the coarse grain fraction. Indeed, such an age underestimation has been reported in the literature, and I do not doubt these findings for the particular sites.

However, in the presented manuscript, only for one out of eight samples, a comparison of both grain size fraction is presented. Both numerical results overlap within uncertainties. All other samples report results, either for the coarse grain (top of the composite profile) or for the fine-grain quartz fraction (lower part). The author's statement on a potential age "underestimation" seems to be trigger by an age inversion in the profile. This age inversion is also present for the fading corrected fine-grain polymineral fraction. Why does this suddenly lead to the conclusion that the fine-grain quartz ages are underestimated (in comparison to the quartz coarse grain ages)? I got the impression that the authors had this idea of a fine-grain quartz underestimation in mind and then tried to see this pattern in their data. It is one possibility that deserves to be discussed, weakly supported by the data though.

We like to clarify that the authors did not go biased into the discussion with 'intent' to find an issue with fine grained quartz. This issue has not even been reported by others for this region. However, we see that the grain size discussion has become redundant and was deleted.

3. Technical detail: The chosen format to apply units is odd, e.g., the authors wrote "471.2±47.1 and 525.8±53.2 ka" instead of "471.2±47.1 ka and 525.8±53.2 ka". There is a general pattern in the manuscript and it should be corrected throughout. If I understand the author guidelines of GChron1 correctly, it should even read, e.g., "(525.8±53.2) ka" because the linked SI brochure refers to the Guide to the expression of uncertainty in measurement; I might be wrong. Besides, I suggest to round values to meaningful digits. If the age uncertainty is around 10°% the number after the digit does not tell much.

1 https://www.geochronology.net/for\_authors/manuscript\_preparation.html

Regarding the three points of this comment:

1. The format of units has been changed according to the referee's suggestion.

2. We have checked the guides and brochures as linked on the GChron webpage but it remains unclear whether ages and uncertainties should be presented in parentheses or not. However,

we have checked previous publications in GChron (esp. 'Highlighted articles') that not seem to have adapted to this style.

3. We agree with the suggestion of using meaningful digits and have rounded values accordingly.

**Detailed comments**

**Abstract**

1. Line 17–18: The statement that the fine-grain quartz ages are underestimated compared to the coarse grain ages does not seem to be supported by the data for three reasons: (1) The authors do not systematically compare coarse grain and fine grain quartz ages, they do this for sample RIN13 only. (2) For this particular sample, coarse and fine grain quartz ages overlap within 1. (3) Table 1 marks the coarse grain age as "underestimated due to grain size-dependent De underestimation", not the fine-grain quartz fraction.

It does not mean that the fine-grain ages are per se not underestimated (e.g., compared the general pattern observed and compared to the profile figure with corrected feldspar ages). Still, the abstract should reflect the essential outcome of the study with regard to the data.

For (1) to (3) see General Comments 2. The abstract has been changed to reflect the essential outcomes.

2. Line 19–20: That the dating reveals a rapid deposition during the (at least) MIS 6 was first mentioned in the Conclusion. Doubtlessly, it had no particular relevance to the authors, given the overall scope, but the abstract leaves the reader with the impression that this point will be detailed in the manuscript.

The context between deposits and chronology has been moved to a subsection and was elaborated on which justifies a mentioning in the abstract.

**Main text**

1. Line 49: "Feldspar" ! "feldspar"

Text has been changed.

2. Line 55: Please report g-values normalised to two days or report the tc value otherwise it will be impossible for readers to compare these values with other findings from the literature.

The g-values reported in line 55 are from the literature as indicated. None of the cited studies do present tc values that might be citable. For completion, we have added the information that only one previous study does present their data normalised to 2 days (Buechi et al., 2017).

3. Line 63: I am not sure whether Thiel et al. (2011) should be mentioned as well here (for the 290 C)? *We agree and the reference has been added.*

4. Line 73: Remove "Scientific"

By stating that a scientific drilling was conducted, it is clear that samples come from drill cores that have been taken for the purpose of scientific investigation and not as a by-product of e.g. a hydraulic rotary drilling that would disturb the integrity of the sediment and may induce further issues that are likely to impact the luminescence signal of the samples.

5. Line 101: ca 106 should suffice.

Text has been changed.

6. Line 110: Add references for the calibration quartz

Manufacturer/provider and batch number of the used calibration quartz are presented. We find this to be sufficient information at this point.

7. Line 115: Better "when comparing", because this is what people do when they double check your protocol parameters.

This is a statement regarding the work of Schmidt et al. and not a general comment. Therefore we would like to keep the text as is.

8. Line 131: "gamma-ray"

Text has been changed.

9. Lines 131–132: There is something wrong with the citation chain:

For the fine grain quartz fraction Buechi et al. (2017, p. 57) wrote: "For fine-grained quartz an a-value of 0.04±0.02 has been incorporated to account for the variability of the values reported in literature (Rees-Jones, 1995; Mauz et al., 2006; Lai et al., 2008).".

For the polymineral fraction Buechi et al. (2017, p. 57) reported: "The effect of alpha irradiation was considered with an a-value of 0.05±0.01 for PM fractions (Preusser, 1999b; Preusser et al., 2001).".

Where Preusser (1999b) is the here cited Preusser (1999). It was Preusser et al. (2001) who reported a-values for the polymineral fraction as quoted in line 132.

Contrary, Preusser (1999) reported four IRSL a-values without uncertainty with a mean of 0.05 and a standard deviation of 0.00 (all values show 0.05; their Table 2). More important is that the applied protocol is not similar to what was applied by the authors here.

In Sec. 4.2.2, the authors detail various possibilities and discuss whether the selected a-value is justified. My impression is that the here applied a-values were used, because they had been always used for samples from that region. This might be justified, but it also shows that it should be remeasured at some point. In either case, the chosen values need the proper reference.

Firstly, the citation chain and references have been fixed.

Secondly, the a-value used for fQ has been changed to  $0.04 \pm 0.02$  following Buechi et al. (2017) thereby incorporating a larger error onto the ages. This will allow to account for at least some of the uncertainty introduced by not measuring out own a-values and keep the results comparable with other studies from the region. While it is certainly favourable to measure study or sample specific a-values, this is not always feasible (i.e. our lab does not have an alpha source).

*Thirdly, the IRSL a-values go all back to Preusser (1999, Kölner Forum für Geologie und Paläontologie, ISSN 1437-3246). Here, the values are reported with uncertainties. Preusser (1999, QSR) and Preusser*

et al. (2001, QR) use these values without providing much further details (these articles focus on reconstructions) and without uncertainties (as tables would not have fitted page width otherwise). The referee is right that the methodology used is different from the once used here. The a-values were determined using the combination of an external alpha (Am-241) and a gamma source (Co-90). While the a-values represents a physical property of a sample material that is independent of how it was determined, one could of cause argue that the lower values are due to a technical issue such as calibration or measurements procedure. However, these values have been determined the same way as several hundreds of values reported in several articles by Manfred Frechen and his team in the 1990th (mainly on loess from Eurasia). For example, Frechen and Preusser (Frankfurter Geowiss. Arb. 20D) and Preusser and Frechen (1999, Terrest. Quartärgeol.) report 25 IRSL a-values with a mean of 0.08±0.01 (hence in agreement with Schmidt et al. 2018). Since these were determined using the same equipment, calibration, procedures and operator, it is likely that the values reported from Switzerland are indeed different (for unidentified reasons).

10. Line 130: U, Th, K concentration values were deduced from gamma-ray spectrometry but only summarised values are presented. What about radioactive disequilibria?

Your environment is undoubtedly very challenging regarding the dose rate, so maybe you can present a few more results regarding the nuclide concentrations? For example, as a plot normalised to Th-232 (cf. Guibert et al., 2009), this would give a good indication. If this appears to be too much, the authors can copy and paste the data from the VKTA into a supplementing document and add one sentence to the main text addressing the possibility of radioactive disequilibria.

A sentence has been added to the main text. As a matter of fact, we regard the environment as not particularly challenging and we regularly check for radioactive disequilibrium. As this is not an issue with our samples, we do not intent to start a discussion on this topic and hence also have not added supplementary information. Upon reasonable request we are happy to hand out data to the interested reader.

11. Line 133: Gaar et al. (2013) confirm Huntley and Baril (1997); which is very reassuring. However, (1) they report  $12.9 \pm 0.4\%$  and (2) they argue for the application of the 95% confidence interval for the potassium concentration (citing Huntley and Baril 1997), means ca  $12.5 \pm 1\%$ . Since the authors cited both references, they should make elaborate why they did not follow the suggestion by Gaar et al. (2013).

**The reference of Gaar et al. has been removed.**

12. Lines 140–180 (Sec. 3.1 Performance test): It should read "tests" and please add further subsections so that the results for the quartz and the feldspar/ polymineral, can be more easily separated.

Text has been changed and the manuscript was substantially restructure allowing to easily find sections for the specific minerals.

13. Line 164: It is not really a different "preheat behaviour" but a completely different design of the heating element and the thermo couple and its feedback electronic. So perhaps: "different technical design"?

Text has been changed to 'reader-specific preheat conditions'.

14. Line 198: "Chinse" -> "Chinese"

Text has been changed.

15. Line 230: The obtained overdispersion value also depends on the initial b; if set. Was it different from zero?

**No, 0 was used.**

16. Line 237–238: I am not sure whether CAM is the most suitable model. The authors should doublecheck the findings by Heydari & Guérin (2018). I also suggest adding one or two dose-response curves from the lower part of the profile.

Certainly, the choice of an appropriate age model is a strongly discussed topic within the luminescence community and consensus is unlikely to be found. We have chosen the CAM out of the large number of models available as most applicable for our datasets, thereby, applying the most commonly used model.

17. Lines 257–258 ("However, if ..."): The De is not a good indication because it is a function of the dose rate and should only be used if the dose rate is homogenous over the profile. Besides, the fading corrected feldspar age appears to be also slightly younger (within 2 ok) for the Kars model. My point: If the authors want to keep that argumentation, they should extend the description of the environmental setting and the dose rate. Ages should not be disconnected from the sedimentological environment. For example, why did not a "facies change" (maybe it is not) cause all these "problems"? *The three sentences have been deleted*.

18. Lines 261: The purple density curve in RIN13 looks somehow skewed.

**All measurements of RIN13 PM are within 1 $\sigma$ of each other.**

19. Lines 269–270: Preusser et al. (2014) wrote that they followed Auclair et al. (2003). Perhaps the latter one is the better reference to cite, or at least in combination with the first. Besides, it appears that Preusser et al. (2014) did not normalise their values to tc as done by Auclair et al. (2003). In that case, the g-value will be slightly different than "expected". The authors may want to add a plot showing their fading measurements; then it should become clear. Additionally, Preusser et al. (2014) measured only three points on the time axis.

The citations have been changed. More details about the fading measurements i.e. storage times, protocol and analysis have been added to the text. With already 13 figures in total, we refrain from adding this information in an illustrated way. 'As the IRSL signal commonly is subject to a loss of signal over time, fading tests were conducted following Auclair et al. (2003) for a given dose of ca. 130 Gy. Three cycles with storage times of ca. 0 s, 1 h, 2.5 h, 5 h, 10 h were implemented per aliquot while the aliquot constantly remained on the sample arm. This reduces the possibility of sample material being lost during mechanical transfer of the aliquot between sample arm and storage carousel (Preusser et al., 2014).'

With regard Kadereit et al. (GChron discussion 2020)2 the obtained g-value might be somewhat arbitrary, and so would be the following fading correction.

Nevertheless, I did not make this a major point for two reasons: (1) The manuscript by Kadereit et al. will likely be rejected and not become published (though the discussion is online and outlines the general problem). (2) Fading measurements and corrections are a tedious business. The approach chosen by the authors might be ok; it might be not. Without further age information, in particular, in the lower part of the composite profile, it is impossible to say.

Firstly, the manuscript referred to has been withdrawn by the authors and can hence not be discussed here. However, we are aware that fading measurements and corrections heavily rely on assumptions and laboratory procedures.

20. Lines 273: Unfortunately, the function the authors applied to corrected the ages after Lamothe et al. (2003) for fading has a (recently discovered) bug (https://github.com/R-Lum/Luminescence/issues/96). The consequence of the bug is that the uncertainty of the fading corrected ages is lower than it should be because the error of the g-value goes into the calculation with a weighting that does not seem to be justified. Of course, this is nothing I hold against the authors. I just wanted to mention it here.

Thank you for mentioning this. It would be good to see uncertainties without the weighting of the g-value, however, this will not change the conclusions drawn from the 'Lamothe' corrected  $D_e$  values.

21. Line 275: Please mention the tc value along with your g-value, otherwise they are not comparable. Please add throughout the manuscript.

To allow for comparison, g-values derived from the presented dataset are reported in % per decade normalised to 2 days. The '2 day#-normalisation has previously only been mentioned in the caption of Table 1. Text has been added to clarify this.

22. Lines 295–298:I was wondering whether the D0 criterion has any substance after the value became corrected for fading after Kars et al. (2008)? With the correction, the De is deduced from a new, simulated, dose-response curve. The D0 of the simulated curve should be the new reference, not the faded dose-response curve. Did I overlook something?

We agree and have reported the simulated  $2*D_0$  values here. However, this has not been properly announced. We have changed the text so that measured and simulated  $2*D_0$  values are presented and distinguishable.

23. Lines 335–336: I do not agree that based on these findings, the logical conclusion is that coarse and fine grain quartz ages are different. The profile may have some hiatus going along with the age inversion. The reason for this age inversion is not necessarily grain-size related. Do the authors have granulometric data from the core?

We apologise but this argumentation is hard to follow. How would a hiatus explain the age inversion with depth? Also, what gain do you expect from quantitatively, in comparison to qualitatively, derived grain size data? Nevertheless, the discussion about grain-size dependent ages has been removed.

24. Lines 340–341: The last point appears like an appendix in this sections and it leads to nothing further. Is this maybe some kind of leftover from a discussion the authors wanted to engage but did not?

The manuscript has substantially been restructure and we have expanded this particular point.

25. Lines 343–365: This is a helpful discussion of different scenarios and justified. However, it should engage a more general discussion on dose-rate scenarios (which does not exist yet).

We don't see the need of a more general discussion at this point; neither evidence for radioactive disequilibrium was found nor do we expect issues from the macro-dosimetry as samples were taken far enough from obvious unit boundaries.

26. Lines 366–389 (4.2.2 Alpha efficiency values and age determination): This subsection renders a potentially fascinating discussion. The problem I have with this section, in particular the first part, is

that it does not read clearly but mixes different aspects. For example, after reading the section, my conclusion was that the chosen a-value of 0.05±0.01 is the less justified value. The reasoning is that somehow all goes back to Preusser (1999) and Preusser et al. (2001) Although for Preusser et al., 2001 I am not sure whether it does not resembles values from Preusser, 1999 (?).

Means, in the worst case, the selection bases on four values with rather low a values, e.g., for the polymineral and feldspar fraction. By contrast, the majority of the other articles would favour higher values. Besides, Schmidt et al. (2018) presented an extended dataset of a-values (IRSL and pIRIR290), though the focus was pIRIR290, which was not measured here.

Of course, it does not mean that the value is wrong, but the arguments presented by the authors indicate that (as even alluded in the manuscript) that they should remeasure the value.

We have added to higher a-value spectrum to Fig. 12 and have discussed its impact on the ages. Also, see comment 9.

27. Line 373–374: Please correct the reference or the a-value (see above)

**References have been corrected.**

28. Line 391: Something is missing in the section title. Perhaps: "Quartz age grainsize dependency" or "Grain-size dependency of the quartz ages"

**We agree, however, the section was removed.**

29. Lines 391–403 (Sec. 4.2.3): The section is very brief and, in my opinion, does not add to the understanding of the "age discrepancies" (if there is any, see comments above) and it does not discuss the quartz grain-size dependency as announced in the section title. Instead, it provides a brief, selective review of other findings, and it concludes that the lowermost two samples should be regarded as minimum ages. I would support the conclusion, but not the reasoning.

**The section has become redundant and was removed.**

**30. Line 400: Timar-Gabor et al. (2017) wrote:**

On the other hand, the age discrepancy of SAR-OSL ages previously reported for Romanian and Serbian loess for ages beyond 40 ka (equivalent doses >100 Gy) was also found to be characteristic of Chinese loess. It is thus believed that this is potentially a global phenomenon, affecting previously-obtained chronologies worldwide, and further increasing concerns for the accuracy of silt-sized SAR-OSL ages in this high dose range. (Timar-Gabor et al., 2017, p. 470).

Timar-Gabor et al. (2017) expressed a guess or hypothesis as part of the conclusion. This conclusion, however, should not be become some statement on the "pattern around the globe'. At least the cited study does not provide the data to it.

Furthermore, Timar-Gabor et al. (2017) refer, first of all, to own observations from Romanian and Serbian loess comparing 4–11  $\mu$ m (fine grain) and 63–90  $\mu$ m (their coarse grain). This is not similar to what is presented in the manuscript for GChron.

**The section has become redundant and was removed.**

31. Lines 411: The conclusion should reflect the results and discussion of the manuscript. The depositional history was not discussed in the manuscript and came here by surprise.

We agree and have changed the conclusion to only report what has been discussed previously.

**Figures and tables**

**1. Figure 1**

• Do the authors have other ice extent data to show? Perhaps the LGM ice extent is a nice to have, but of limited relevance given the age results.

The LGM ice extent is presented to emphasis the fact that no glacial overprint has occurred during 'more recent' times. The only other extents that might be reasonable to be shown are the ones of the Möhlin and Beringen glaciations. The maximum extent of these glaciations are situated much further to the North within Germany and would thereby cover the entire extent of the area presented in Fig. 1. We do not believe that adding this extra information to Fig. 1 will contribute to the understanding of the here presented study. However, text has been added to subheading '2 Site setting and samples'.

• "A." and "B." is part of the figure, consequently those lettering should be part of the figure caption.

The figure caption has been changed accordingly.

- 2. Figure 2
  - An y-axis unit (core depth) is missing.

The unit (m) has been added.

• I would be good to have some photos showing the core log to better understand the setting. Also, the authors may want to indicate where one core ends and the next starts.

This goes beyond the scope of this figure and this study. The full core log and photos are available online. The reference is given in subheading 2 (i.e. Gegg et al., 2018), however, a link to the webpage has been added to the references (https://www.nagra.ch/de/cat/publikationen/arbeitsberichte-nabs/nabs-2018/downloadcenter.htm).

• What does the upper x-axis ("C, Si, Sa, Gr, Co, Bo" on top the core log) labels? Probably it is obvious, but it is not to me and maybe other readers are not familiar with it as well.

The upper x-axis presents the dominant grain size as presented in the log units. For explanation, text has been added to the caption.

• The age comparison should be based on 95% confidence intervals. However, I guess the graph will not scale very nicely given the two lower polymineral ages. This can be fixed by using a non-continuous scale.

*This would be one way of presenting the data, we have chosen not to. By introducing discontinuous x-axes, the figure will become messy and gets less easily understandable.*

**3. Figure 3**

• The figure would benefit from more details in the figure caption. Readers not familiar with luminescence dating may struggle to understand the figures. For example: "M/G" probably means measured to given dose, "PH" means preheat etc.

Text has been added to the caption to explain 'M/G' while the x-axes labels have been adjusted to 'Preheat temperature' to avoid unnecessary abbreviations.

• The inset legend in all the figures in the right column is unnecessary because only one type of data is shown in all figures. It would suffice if the figure title (or subtitle) says "thermal transfer test". The way the figures are presented are consistent and will allow the reader to easily comprehend the different tests presented here.

• It is not clear what the data points are displaying. A single measurement with uncertainties? An average with the error bars showing the standard deviation (of the mean)?

Presented are CAM  $D_e$  values with 1  $\sigma$  uncertainties. Text has been added to the caption and the number of measurements are mentioned for clarification.

**4. Figure 4**

• The mineral fraction is missing in the figure caption

The mineral fraction 'F' has been added to the caption.

**5. Figure 5**

• Same as above, the mineral fraction is missing in the figure caption

The mineral fraction 'F' has been added to the caption.

**6. Figure 6**

Assuming this is referring to Fig. 7:

• Same problem as Fig. 3. The figure captions should explain used abbreviations (e.g., "DR")

*Explanation for used abbreviations has been added to the caption.*

• The solid line the curve is not really showing the "given dose decay" but the "luminescence signal decay" or shine-down curve of the natural signal. It is a proxy for the "given dose decay", but is not a "dose decay".

**Text has been changed to 'luminescence signals of the natural or first given dose' in the caption and the text in the figure has been adjusted accordingly.**

• "Natural decay" might lead to a wrong understanding by others who work with dating methods relying on the "radioactive decay" of isotopes. Perhaps: "natural shine-down curve" or something similar.

**See above.**

• De(t) plot was used by Bailey et al. (2003) to identify the partial resetting of the luminescence signal. They shifted and extended the signal integrals slightly at the end. Probably it was not done here, but the figure caption should detailed what was done so that the figure becomes immediately understandable.

*Text has been added to the caption to clarify that 0.4 s intervals (Q, fQ) and 1.5 s (F, PM) intervals were used.*

• Y-axis labelling should be added to the figure in the right column.

We believe this is redundant and will make the figure messy as not only y-axis labels would have to be added to the primary y-axis of the right column but also to the secondary y-axis of the left column.

7. Figure 12

• Proper x-axis labelling is missing or figures should align more closely.

Labels have been added to all x-axes.

• Was a similar bandwidth used for all kernel density curves?

Bandwidth was determined for each *D*e distribution individually.

• RIN2: Table 1 reads 158.4±4.4 Gy instead of 158.4±4.3 Gy in the figure (minor detail, since I argue for meaningful rounding above).

Text has been adjusted in Fig. 12 and numbers are now presented as integers.

8. Figure 13

• 95% confidence intervals should be used for the age comparison.

Confidence intervals are one way of presenting an age comparison. We have decided to take another way and only show the central age values to keep the figure clear and understandable.

• There is somehow a typo in the figure caption: It reads "Accepted ages are presented with 1 uncertainty as point symbol.", however, there is no "point symbol" in the figures.

Text has been changed to clarify that this referring to signature styles.

9. Table 2

• What meant is the internal K-concentration, it should be written.

Text has been changed accordingly.

• "De" should read "De" (subscript "e")

Text has been changed.

**Personal note to the authors**

Dear Müller et al.,

I can imagine that you do not agree with my suggestion to transfer the manuscript. To avoid the impression that I am "against" your manuscript, I may add that I sincerely believe that every luminescence-dating study deserves to be published; given that it is free of significant mistakes. Luminescence dating is way too costly to ignore the data or let them disappear in a drawer. Your study indeed, should be published. However, I think for GChron it would need way more tests (e.g., TR-OSL, TL with trap parameters etc.) and a larger dataset. This is nothing I can ask of you. To the contrary, you probably have the perfect for, e.g., E & G with a ready to go chronological part if you extent the geomorphological/geological part.

Nonetheless, to ease the minds and to avoid a heated discussion: If the editor believes that the manuscript is most suitable for GChron, I will certainly not further argue against a publication in GChron. Moreover, I consider my suggestions as a piece to an open discussion, which is not carved in stone.

Thank you for clarifying your intentions and your thorough review of the manuscript. For a detailed response to the raised concerns regarding the manuscripts suitability for GChron and the suggestion to transfer the manuscript to a different journal see above.

**Conflict of interest**

I have no conflict of interest to declare. I am not a beneficiary of the suggested references to be cited. Naturally, the authors are free to reject my reference suggestions.

**Response to Associate Editor Decision for GChron-2020-15**

We would like to thank the Associate Editor, Julie Durcan, for her decision to publish our manuscript in Geochronology after minor revisions and for the constructive comments she made. We agree with most comments and have added or adjusted text accordingly. Below we respond to her comments in detail.

**Comments to the Author**

In this paper by Mueller and colleagues, the authors address the dating of glacial and periglacial sediments in the Alps. As is the case in many glacial contexts, the dating of peri-/glacial sediments has been reported as problematic in terms of luminescence dating. This paper provides the detailed testing of coarse grain quartz and feldspar and fine grain quartz and polymineral luminescence signals in an attempt to resolve phases of glacial advance in the northern foreland of the Swiss Alps. Whilst the paper identifies valley fill in the area at least back to MIS6, discrepancies between the various dating signals are identified and as yet remain unresolved.

I am satisfied that this paper goes beyond a 'routine' dating study (and I echo Sebastian Kreutzer's comments that there is nothing at all wrong with routine dating studies), and therefore I believe it suitable for publication in Geochronology. This paper demonstrates that luminescence dating is more than an 'off-the-shelf' dating technique, and that efforts to interrogate the signals being dated are key in developing chronologies in geomorphic contexts which traditionally have been a challenge for luminescence daters.

I thank both reviewers for their detailed reviewers, and believe that the authors have provided a satisfactory response to provide an improved manuscript. There are a couple of comments in Sebastian Kreutzer's review which I agree are still relevant, although the authors haven't modified significantly their manuscript in response to these. These are i) the difficulties in undertaking fading measurements and analysing the subsequent data – this is where we are with luminescence dating currently, but it is important to acknowledge this; ii) the use of the central age model – and I highly recommend the paper of Heydari and Guerin (2018) to the authors which explores the use of the average dose model (Guerin et al., 2017). Whilst CAM is one of the most (if not the most) commonly used age models in luminescence dating, it shouldn't be immune from supersedure; iii) acknowledgement of the complexities of dose rate calculation, in particular issues arising from disequilibrium if present and undetected. I highlight these in order that the authors may consider these issues in future research.

Thank you for highlighting these points, we certainly will consider them in our future research. Also, we would like to make some finally comments to these points:

i) Fading measurements come with a suite of uncertainties and issues themselves. To acknowledge this, we have added the following sentence under the fading implication subsection (6.2): [..] however, it is recognizing that fading measurements and corrections come with a suite of uncertainties.

ii) We agree, the CAM is not the 'ultimate' age model that is immune to supersedure. In case of the here presented data, application of e.g. the Average Dose Model (ADM) will not make a significant difference. We have plotted (uncorrected) CAM De values against those of the ADM to emphasise this argument (see below). Except for one sample, all results are well within 1 σ. The exception is sample RIN13 Q that gives De values just within 1 σ. For this samples and OD of 56 % was reported and only a minimum age estimate was obtained.

iii) Dose rate determination is indeed a complex endeavour and we are aware that the detection and monitoring of disequilibria are an essential part for the establishment of accurate chronologies. Therefore, all our samples are measured routinely using high-resolution gamma spectrometry and results are checked for disequilibria. We have added a remark regarding this to the manuscript.

Further to the reviewer's comments, I have a few minor comments:

Abstract: Please expand the abstract to summarise the suite of tests you undertook on quartz and feldspar in order to provide a chronology. At present, the abstract reads like a 'regular' dating paper where you used a number of different signals, and your paper goes beyond this.

We agree and have emphasised this point accordingly.

L12: in order to overcome what? E.g. to provide reliable chronologies

We agree and the text has been changed accordingly.

L98: 'deposited' is probably more appropriate for 'emplaced' – assuming that's what you mean.

**We agree and the text has been changed.**

L204: Please can you clarify whether your EMCCD measurements made after lab irradiation. As a comment only, I'm not surprised your detection of signals is so low, given the low sensitivity of the EMCCD system (for Riso systems at least- see Thomsen et al, 2015 DOI: 10.1016/j.radmeas.2015.02.015)

EMCCD measurements were conducted on the natural signals which has been clarified in the text. In addition, the EMCCD system was specified under subsection 3. (Sample preparation, equipment and dose rate determination). We also want to highlight that the observed low quartz seinsitivty is typical for the region (e.g. Trauerstein et al., 2017) and not necessarily solely explained by the use an EMCCD system.

L258: Please can you provide a short sentence to summarise the key differences between the two fading procedures.

The following sentence has been added: The first approach is based on the extrapolation of luminescence signal loss to a decadal percentage (g-value) that is used to correct the natural dose and the dose response points while the latter approach utilises the sample specific density of recombination centres ( $\rho'$ -value) to account to correct the dose response curves.

L444: For me, that not all minerals/fractions/samples are suitable for dating demonstrates the strength of your approach of systematically and testing in detail a range of signals. This is worth commenting on in your conclusion

We agree and have emphasised this point in the conclusion by adjusting the text: Not all minerals, fractions or samples are suitable to be dated using the here presented luminescence techniques, nonetheless, this systematic investigation of different luminescence signals allows for a decisive chronology to be established.

Fig 10 caption. Can you add whether these are signals from natural or lab-irradiated signals (if so, what dose please)? Also, the size of the ROI?

Text has been added to the caption to clarify that EMCCD measurements were conducted on the natural signals and that holes (ROI) are 300  $\mu$ m in diameter.

**Luminescence properties and dating of **pro**glacial to periglacial sediments from northern Switzerland**

Daniela Mueller1, Frank Preusser1, Marius W. Buechi2, Lukas Gegg2, Gaudenz Deplazes3

1Institute of Earth and Environmental Sciences, University of Freiburg, 79104 Freiburg, Germany
 2Institute of Geological Sciences, University of Bern, 3012 Bern, Switzerland
 3NAGRA - Nationale Genossenschaft für die Lagerung radioaktiver Abfälle, 5430 Wettingen, Switzerland

Correspondence to: Daniela Mueller (daniela.mueller@geologie.uni-freiburg.de)

- 10 Abstract. Luminescence dating has become a pillar of the understanding of Pleistocene glacial advances in the northern foreland of the Swiss Alps. However, both quartz and feldspar from the region are equally challenging as dosimeters with anomalous fading and, partial bleaching and unstable components being some of the obstacles to overcome for thete establishment of decisive-reliable chronologies. In this study, luminescence properties of coarse- and fine-grained quartz, feldspar and polymineral fractions of eight samples from a palaeovalley, Rinikerfeld, in northern Switzerland are
- 15 systematically assessed. Standard performance tests are conducted on all four fractions. Deconvolution of luminescence signals of the quartz fractions is implemented and shows the dominance of stable fast components. Reader specific low preheat temperatures are investigate on the infrared stimulated luminescence (IRSL) signal of feldspar. Thermal stability of this signal is found for low preheats and thermal quenching could be excluded for higher preheats. -and found appropriate for dating. WhileHowever, anomalous fading is observed of the IRSL signal of in the feldspar and polymineral fractioninfraIRSL red

[revised manuscript text omitted]